# Fast uncovering of protein sequence diversity from structure
# Conference Submissions

**Luca Alessandro Silva**
Department of Computing Sciences
Bocconi University
Milan, MI 20100, Italy
`silva.luca@phd.unibocconi.it`

**Barthelemy Meynard-Piganeau**
Institut de Biologie Paris Seine, Biologie Computationnelle
et Quantitative LCQB
Sorbonne Université
Paris, France
`barthelemy.meynard@polytechnique.edu`

**Carlo Lucibello**
Department of Computing Sciences
BIDSA
Bocconi University
Milan, MI 20100, Italy
`carlo.lucibello@unibocconi.it`

**Christoph Feinauer**
Department of Computing Sciences
Bocconi University
Milan, MI 20100, Italy
`christoph.feinauer@unibocconi.it`

## Abstract

We present InvMSAFold, an inverse folding method for generating protein sequences that is optimized for diversity and speed. For a given structure, InvMSAFold generates the parameters of a probability distribution over the space of sequences with pairwise interactions, capturing the amino acid covariances observed in Multiple Sequence Alignments (MSA) of homologous proteins. This allows for the efficient generation of highly diverse protein sequences while preserving structural and functional integrity. We show that this increased diversity in sampled sequences translates into greater variability in biochemical properties, highlighting the exciting potential of our method for applications such as protein design. The orders of magnitude improvement in sampling speed compared to existing methods unlocks new possibilities for high-throughput virtual screening.

## 1 Introduction

Inverse folding aims to predict amino acid sequences that fold into a given protein structure, and plays a fundamental role, for example, in the protein design pipeline of RFDiffusion (Watson et al., 2023). Recent deep learning approaches such as ESM-IF1 (Hsu et al., 2022) or ProteinMPNN (Dauparas et al., 2022) achieve remarkable accuracy in this task. However, instead of predicting a single ground truth sequence, it is often desirable to have a method that is able to generate a variety of different sequences with the desired fold, i.e., solving a *one-to-may* problem, see Fig. 2. This diversity could be leveraged for example by starting from a source sequence Sturmfels et al. (2022); Bryant et al. (2021) and taking different molecular environments into consideration (Krapp et al., 2023). Such an approach would expand the sequence design space while preserving structural consistency, enabling a larger pool of sequences for selection based on additional properties like thermostability, solubility, or toxicity. In drug discovery, for example, it would facilitate the generation of a large number of diverse candidates, allowing further selection optimized for properties such as bioavailability. Similarly, in biotechnology and enzyme engineering, it would facilitate the creation of enzymes with tailored properties, such as improved stability and activity under varying conditions. On the computational side instead, even after training, sampling from transformer-based architectures such as ESM-IF1 Hsu et al. (2022) or ProteinMPNN Dauparas et al. (2022) can be very expensive. This can severely limit the widespread use of such models, especially in virtual screening-like settings.

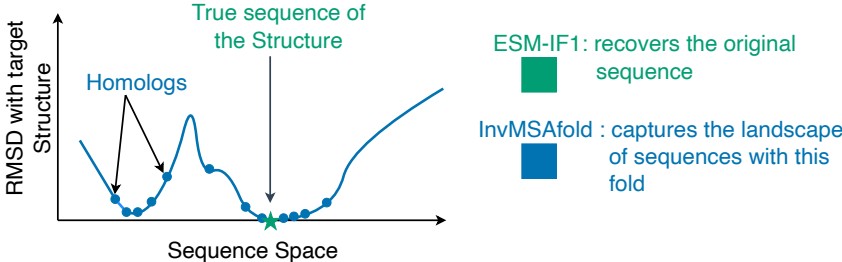

Figure 1: InvMSAfolds expands the scope of inverse folding to the retrieval of the entire landscape of homologous proteins with similar folds.

In this work, we present an efficient method that is able to generate diverse protein sequences given a structure, including sequences far away from the native one (see Fig. 1). Recent architectures for inverse folding are based on encoder-decoder architectures, where a structure is encoded and a sequence decoded. During training, such models typically take into account only the native sequence of a given structure, maximizing its probability given the structure (Hsu et al., 2022; Dauparas et al., 2022).

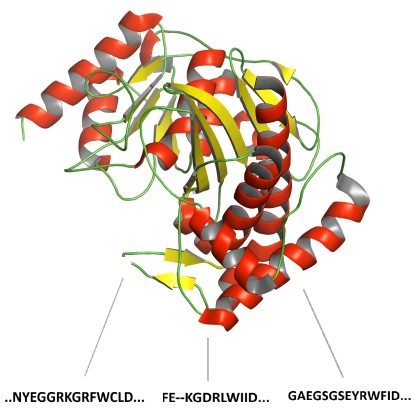

Figure 2: *One-to-many* nature of inverse folding. **Top**: Structure of 1KA0. **Bottom**: Some homologos sharing the fold.

In our approach, we use the decoder to generate the complete set of parameters for a lightweight model that is sufficiently expressive to describe the sequence diversity of the *multiple sequence alignment* (MSA) of the native sequence. This architecture is trained end-to-end to capture the broader probability distribution of sequences corresponding to a specific fold. We consider different choices for the lightweight model, and settle on the pairwise family (Figliuzzi et al., 2018), since it has been widely applied to protein sequence data Cocco et al. (2018a), proven to be good at generation (Russ et al., 2020) and demonstrated to capture information that enables fitness prediction (Poelwijk et al., 2017). A potential issue with pairwise models is that they typically have a number of parameters that is quadratic in the sequence length. We solve this issue by considering low-rank approximations, thus reducing the effective number of parameters to the same order of the sequence length. By doing so we drastically reduce the number of parameters, boosting training efficiency.

Our model generates all of these parameters in a single forward pass, similar to previous research Li et al. (2023). Once this generation is done, the resulting pairwise model can be used for generating a large number of diverse sequences very efficiently leveraging CPUs on a standard machine, dramatically facilitating its use. We show that the models we generate are able to capture the diversity of the protein family better than other models and are able to find sequences far away from the natural sequence that are predicted to still fold into the same structure. We also show that this increased diversity translates into a more spread distribution in other properties, enabling selection of promising sequences from a larger pool. Codes to train the models and replicate some of the results can be found at the Potts Inverse Folding repository.

## 2 METHODS

In this Section, we describe the components of our architecture, InvMSAFold, and its training procedure. Given a structure-sequence pairing $(\boldsymbol{X}, \boldsymbol{\sigma}_X)$, inverse folding methods typically define a probability distribution on the space of sequences $\boldsymbol{\sigma}$ as $p(\boldsymbol{\sigma}|\boldsymbol{X})$. Most deep learning-based methods

model this distribution auto-regressively, using $p(\boldsymbol{\sigma}|\boldsymbol{X}) = \prod_{i=1}^{L} p(\sigma_i|\sigma_{i-1}, \ldots, \sigma_1, \boldsymbol{X})$, where $L$ is the length, and train by minimizing the loss on the true sequence $\boldsymbol{\sigma}_X$, possibly after adding noise to the coordinates $\boldsymbol{X}$ (Hsu et al., 2022; Dauparas et al., 2022). Sampling amino acids auto-regressively requires a full forward pass through the neural network for every generated amino acid, making it very expensive to use in a virtual screening-like setting, where a large number of sequences are scanned for properties beyond folding into structure $\boldsymbol{X}$. Moreover, minimizing the loss on the true sequence ignores the *one-to-many* property of inverse folding depicted in Figure 2, resulting in a distribution peaked around very few sequences (as we will show later) and not capturing other parts of sequence space that might be interesting for the problem at hand.

We address both issues, reduced diversity and slow sampling, by having InvMSAFold output a simple and easy-to-sample-from model for fast inference and by training our architecture over multiple sequence alignments for any given input structure instead of a single sequence.

## 2.1 THE INVMSAFOLD ARCHITECTURE

InvMSAFold is a neural network whose inputs are the structure backbone coordinates $\boldsymbol{X}$ and whose outputs are the parameters of a lightweight sequence model. These parameters, $\boldsymbol{\theta}(\boldsymbol{X})$, are then used to sample amino acid sequences compatible with the given structure. Therefore, our overall model takes the form $p(\boldsymbol{\sigma}|\boldsymbol{X}) = p(\boldsymbol{\sigma}|\boldsymbol{\theta}(\boldsymbol{X}))$.

We explore different parametric families for the lightweight model. In order to go beyond distributions that treat different positions in the protein as independent and to ensure sufficient expressivity, we focus on pairwise models Cocco et al. (2018a) that correspond to the well-known Potts model (Barrat-Charlaix, 2018) from statistical physics. Such models have an experimentally validated ability to capture structure-sequence relationships Russ et al. (2020). We therefore train the neural network to learn the mapping $\boldsymbol{X} \rightarrow \boldsymbol{\theta} = (\mathbf{J}, \boldsymbol{H})$, where $\mathbf{J}$ is a tensor of size $L \times q \times L \times q$ and $\boldsymbol{H}$ is a matrix of size $L \times q$, with $L$ the sequence length and $q$ the number of different amino acids. As is common in pairwise models, we call the quantities $H_{i,a}$ the *fields*, indexed by position $i$ and amino acid type $a$, and the quantities $J_{i,a,j,b}$ the *couplings*, indexed by a pair of positions $i, j$ and a pair of amino acid types $a, b$. Fields describe the propensity of amino acids to appear at given positions, while couplings describe the propensity of pairs of amino acids to appear at pairs of positions.

The complete InvMSAFold architecture is composed of two parts, the structure encoder and a decoder which outputs the fields and couplings, see Fig. 3.

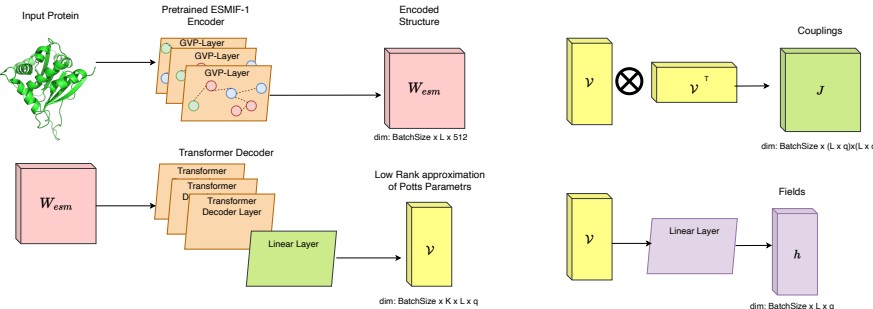

Figure 3: **Left**: Decoder architecture up to the generation of the low-rank tensor $\mathbf{V}$. Right: The low-rank tensor $\mathbf{V}$ is used to generate the couplings $\mathbf{J}$ and the fields $\boldsymbol{H}$.

For the encoder, we use the pre-trained encoder from the ESM-IF1 model of Hsu et al. (2022) which follows the GVP-GNN architecture proposed in (Jing et al., 2020). This encoder gives a rotationally invariant representation of the input $\boldsymbol{X}$ that has been shown to be effective in capturing the geometric features coming from the protein's 3D structure. During training we add Gaussian noise to these representations as commonly done in previous research Hsu et al. (2022). The decoder takes as input a batch of encoded structures, padded to a common length $L$. These $L \times D$ matrices ($D = 512$ in our experiments) batched into a tensor are passed through 6 Transformer layers with 8 attention heads. The output is first embedded into a $L \times DK$ dimensional space through a linear layer followed by an

element-wise ReLU activation function, then it is reshaped to a $L \times K \times D$ tensor and finally it is projected to a $L \times K \times q$ tensor $\mathbf{V}$ through another linear layer. By selecting $K \ll L$, we then create from $\mathbf{V}$ a low-rank coupling tensor according to

$$J_{i,a,j,b} = \frac{1}{\sqrt{K}} \sum_{k=1}^{K} V_{i,k,a} V_{j,k,b}. \tag{1}$$

Thanks to this low-rank structure, we drastically reduce the number of parameters of our pairwise model from $L^2 \times q^2$ to $L \times q \times K$. The $1/\sqrt{K}$ factor instead is used to obtain $\mathcal{O}(1)$ at initialization, similar to scaled dot product attention in Vaswani et al. (2017). We note that in other settings, low-rank decompositions like equation 1 of $\mathbf{J}$ have been shown to be similarly effective for most tasks as their full-rank counterparts (Cocco et al., 2013). The fields $\boldsymbol{H}$ are computed by passing $\mathbf{V}$ through another linear layer, and then contracting the tensor by summing the entries across the latent dimension. Also here after summing we scale by $\sqrt{K}$ to ensure the fields to remain $\mathcal{O}(1)$.

## 2.2 INVMSAFOLD-PW

Given the parameters $\mathbf{J}$ and $\boldsymbol{H}$ we explore two different approaches to define a probability distribution over sequences of amino acids. The first distribution, which we term *InvMSAFold-PW*, is a standard pairwise distribution Cocco et al. (2018a) that defines the negative log-likelihood of a sequence $\boldsymbol{\sigma}$ as

$$\log p^{pw}(\boldsymbol{\sigma}|\boldsymbol{H}, \mathbf{J}) = -E(\boldsymbol{\sigma}) - \log(Z^{pw}) = \left( \sum_{i<j}^{L} J_{i,\sigma_i,j,\sigma_j} + \sum_i H_{i,\sigma_i} \right) - \log(Z^{pw}), \tag{2}$$

where $Z^{pw}$ is a $\boldsymbol{\sigma}$-independent normalizing constant and $E(\boldsymbol{\sigma})$ is called *energy*. This distribution has been explored extensively for proteins Cocco et al. (2013) as it possesses some nice theoretical properties Barrat-Charlaix (2018), yet has the disadvantage that $Z^{pw}$ is intractable, making likelihood optimization difficult. To overcome this issue, a standard alternative is to resort to pseudo-loglikelihood (Besag, 1977; Ekeberg et al., 2013)

$$p\mathcal{L}(\boldsymbol{\sigma}|\boldsymbol{H}, \mathbf{J}) = \frac{1}{L} \sum_{i=1}^{L} \log p^{pw}(\sigma_i|\boldsymbol{\sigma}_{\backslash i}, \boldsymbol{H}, \mathbf{J}), \tag{3}$$

which is a statistically consistent estimator Barrat-Charlaix (2018). Moreover, as we will show, pairing Eq. 3 with the low-rank assumption on the couplings (Eq. 1), allows for very efficient computations.

### 2.2.1 FAST PSEUDO-LIKELIHOOD COMPUTATION

By enforcing the low-rank constraint on the matrix $\mathbf{J}$, the number of parameters is reduced from $\mathcal{O}(L^2)$ to $\mathcal{O}(L)$. However, calculating the pseudo-likelihood naively by materializing the full coupling matrix $\mathbf{J}$ as in Eq. 1 leads to a quadratic computational cost. To achieve linear cost in both memory and computational complexity we note that, given the low-rank structure, the coupling term in Eq. 3 becomes

$$\sum_{i<j,a,b} J_{i,\sigma_i,j,\sigma_j} = \frac{1}{2} \sum_{k=1}^{K} \left( \sum_{i=1}^{L} V_{i,k,\sigma_i} \right)^2 - \frac{1}{2} \sum_{k=1}^{K} \sum_{i=1}^{L} V_{i,k,\sigma_i}^2, \tag{4}$$

which can be computed in $\mathcal{O}(L)$ time since it contains a single sum over the positions $i$. For position $p$, the pseudo-likelihood is

$$p^{pw}(\sigma_p|\boldsymbol{\sigma}_{\backslash p}, \boldsymbol{H}, \mathbf{J}) = \frac{p^{pw}(\boldsymbol{\sigma}|\boldsymbol{H}, \mathbf{J})}{\sum_{c=1}^{q} p^{pw}(\sigma_p = c, \boldsymbol{\sigma}_{\backslash p}, \boldsymbol{H}, \mathbf{J})} = \frac{\exp\{-E(\boldsymbol{\sigma})\}}{\sum_{c=1}^{q} \exp\{-E(\sigma_p = c, \boldsymbol{\sigma}_{\backslash p})\}}, \tag{5}$$

where the numerator is independent of $p$ and therefore the same for all positions. If we can compute the denominators efficiently for every position $p$, we can then maintain a linear cost also for the sum of Eq. 3. Let's call $\tilde{\boldsymbol{\sigma}}^{p,c}$ the configuration that is equal to $\boldsymbol{\sigma}$ except for having amino acid $c$ in position $p$. The relevant quantity for the computation of the energy $E(\tilde{\boldsymbol{\sigma}}^{p,c})$ can be written as

$$\sum_i V_{i,k,\tilde{\sigma}_i^{p,c}} = \sum_i V_{i,k,\sigma_i} - V_{p,k,\sigma_p} + V_{p,k,c}. \tag{6}$$

The first term is common to all positions $p$, hence can be computed just once. Combining Eq. 6 and Eq. 4 we can compute all the denominator terms in Eq. 5 for the different positions $p$ efficiently, resulting in $\mathcal{O}(L)$ cost for Eq. 3. Finally, for the quadratic regularization on the couplings we have

$$\sum_{i<j}\sum_{a,b} J_{i,a,j,b}^2 = \frac{1}{2}\sum_{k,k'}\left(\sum_{i,a} V_{i,k,a}V_{i,k',a}\right)^2 - \frac{1}{2}\sum_{k,k'}\left(\sum_{i}\left(\sum_a V_{i,k,a}V_{i,k',a}\right)^2\right), \quad (7)$$

which leads again to linear cost. Similar reasoning can be applied to the field-dependent part of the energy.

## 2.3 INVMSAFOLD-AR

Although, as previously mentioned, the pseudo-log-likelihood Eq. 3 is a statistically consistent estimator, it is known in practice it could struggle to fit the second moments of the MSAs it is trained on Trinquier et al. (2021). Moreover, for sampling, we have to resort to MCMC algorithms, which can have difficulties in navigating the complex high-dimensional landscape given by Eq. 2.

Based on the above practical limitations, we consider an alternative efficient auto-regressive variant of pairwise models Trinquier et al. (2021), which we term InvMSAFold-AR, that defines an autoregressive distribution $p^{ar}$ over amino acids having negative log-likelihood

$$\log p^{ar}(\sigma_i | \sigma_1, \ldots, \sigma_{i-1}, \boldsymbol{H}, \boldsymbol{\mathsf{J}}) \propto H_{i,\sigma_i} + \sum_{j=1}^{i-1} J_{i,\sigma_i,j,\sigma_j}. \quad (8)$$

Since this parametrization decouples into a sequence of univariate distributions, it has the advantage that the normalization factor is tractable allowing for closed-form computation of the likelihood. As a result, we can train with *maximum likelihood* and also sample much more robustly from the model. Since for InvMSAFold-AR the number of terms in the sum in Eq. 8 depends on the position $i$, we rescale the couplings as

$$J_{i,a,j,b} \leftarrow \frac{J_{i,a,j,b}}{max(i,j)}, \quad (9)$$

following what was done in (Ciarella et al., 2023). We found this to be beneficial for training and note that without this scaling the neural network would have to generate couplings of significantly different magnitudes for different sites. This would not be a problem if we were to optimize the couplings directly, as in previous research Trinquier et al. (2021), but might be problematic if the couplings are generated by a neural network.

## 2.4 TRAINING ON HOMOLOGOUS SEQUENCES

In most other works Hsu et al. (2022); Dauparas et al. (2022), inverse folding models are trained to predict the ground truth sequences only. In this work, we aim to generate a distribution that captures the complete sequence space that is compatible with the input structure $\boldsymbol{X}$. To this end, we leverage the ground truth sequence corresponding to a structure $\boldsymbol{X}$ to extract an MSA $\boldsymbol{M}_X$ from a sequence database (see next section for details). We then use the pairs $(\boldsymbol{X}, \boldsymbol{M}_X)$ for training by taking the mean negative (pseudo) log-likelihood of a random subsample of sequences in $\boldsymbol{M_X}$ given the parameters $\boldsymbol{\theta}(\boldsymbol{X})$ generated by the network. For both InvMSAFold-PW and InvMSAFold-AR, we add a regularizing $L_2$ term for the fields and the couplings as is typical for these models Barrat-Charlaix (2018), Cocco et al. (2018b). For details, see Sec. 4.1.

## 3 DATA AND TRAIN-TEST SPLITS

In order to control the level of homology in our evaluation, we create three test sets, which we call respectively *inter-cluster*, *intra-cluster*, and *MSA* test set. We base these on the CATH database Sillitoe et al. (2021), which classifies protein domains into superfamilies and then further into clusters based on sequence homology. We use the non-redundant dataset of domains at 40% similarity and associate to every domain a cluster as indicated in the CATH database. We then choose 10% of the

sequence clusters uniformly at random and assign them to the inter-cluster test set, excluding these clusters from the training set. Because many superfamilies contain only one sequence cluster, there is a significant amount of superfamilies that appear only in the inter-cluster test set and not in the training set, making this a hard test set. We then create the less stringent intra-cluster test set by taking from every domain sequence cluster that is not in the inter-cluster test set and has at least two domains a single random domain. We then use the remaining domains as the training set. Finally, we create MSAs for all sequences in the datasets using the MMseqs2 software and the Uniprot50 database. We further split the sequences in the MSAs into 90% used for training and 10% for the MSA test set. This last test set is the least stringent one since it is based on domains that are also in the training set. The resulting sizes of the datasets are the following: 22468 for the training set, 22428 for the MSA, 1374 for the *intra-cluster*, and 2673 for *inter-cluster* test sets respectively.

## 4    RESULTS

### 4.1    MODEL TRAINING

For both training and inference, we create embeddings for the structures using the ESM-IF1 encoder (Hsu et al., 2022). We then add independent Gaussian noise with standard deviation equal to $5\%$ of the standard deviation of the embeddings across all positions, dimensions, and samples in the training set. The random MSA subsets $M_X$ are independently generated at each training step, where the number of sequences sampled is a model hyperparameter. For InvMSAFold-PW, we train with a single structure in each batch, with a MSA subsample size for $M_X$ of $64$, a rank $K$ of $48$, a learning rate of $10^{-4}$ and L2 regularization constants of $\lambda_h = \lambda_J = 10^{-4}$ for fields and couplings. For InvMSAFold-AR, we tune the hyperparameters as discussed in Appendix A.2.2 for the details. Both models are trained with AdamW optimizer for a total of $94$ epochs. We monitor the negative pseudo-loglikelihood for InvMSAFold-PW and the negative likelihood for InvMSAFold-AR on the train and the different test sets. Training and test curves are reported in Appendix A.2.1, here we just mention that the ordering of the losses on the different datasets is consistent with the hardness reasoning behind the split in the last section, yet all curves are significantly better than a null model, signaling that the model is able to generalize in all cases.

### 4.2    SYNTHETIC DATA GENERATION

Transformer-based architectures such as ESM-IF1 (Hsu et al., 2022) or ProteinMPNN (Dauparas et al., 2022) can be very expensive to use at inference time, as data generation requires a forward pass through the decoder for every sample. On the other hand, InvMSAFold requires just a single forward pass through the decoder after training; once the parameters $\mathbf{J}$, $\boldsymbol{H}$ of the pairwise model are obtained (either of the two parametrizations), it can be run on the CPUs of a standard machine to produce many samples. The sampling speed is drastically faster than ESM-IF1 or ProteinMPNN.

As can be seen from Figure 4, the sampling speed on CPU of InvMSAFold-AR is orders of magnitudes faster than the one of ESM-IF1 on GPU (notice that the $y$-axis is on log-scale). Moreover, given the lightweight model generated by InvMSAFold-AR, and the fact that host RAM is typically more abundant than GPU memory, sampling for InvMSAFold-AR can be easily batched, resulting in a virtually constant sampling time across all lengths $L$. This is crucial in a virtual screening-like setting, where proteins in the order of millions are generated and analyzed for properties beyond folding into a desired structure; looking at Figure 4, this would be unfeasible for ESM-IF1. In Figure 4, we report just InvMSAFold-AR and not InvMSAFold-PW, as the latter requires an MCMC sampler, while the former and ESM-IF1 return i.i.d. samples from their respective distribution. To sample from ESM-IF1 we utilized a NVIDIA GeForce RTX 4060 Laptop having 8Gb of memory, while for InvMSAFold-AR we used a *single* core of a i9-13905H processor.

### 4.3    COVARIANCE RECONSTRUCTION

The ability of a generative model to reproduce covariances between amino acids in MSAs has been shown to be a good metric for measuring how well the protein landscape is captured (Figliuzzi et al., 2018). Models with this ability have been shown to enable efficient sampling of experimentally validated, functional protein sequences (Russ et al., 2020). In order to test this ability, we created

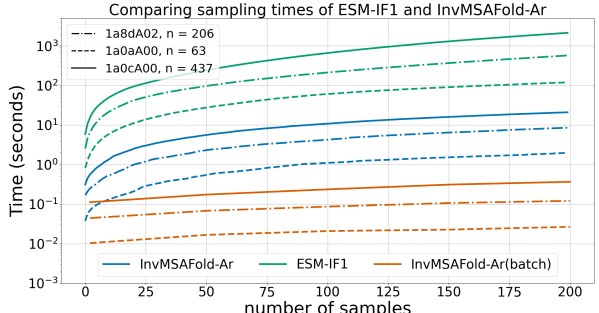

Figure 4: Comparing the sampling speeds in seconds ($y$-axis in logscale) of ESM-IF1 (green) and InvMSAFold-AR (blue non-batched, red batched) for 1a0aA00, 1a8dA02, 1a0cA00 having increasing lengths $L$ respectively of $63, 206, 437$ as we increase the number of samples generated ($x$-axis).

|  | ESM-IF1 | InvMSAFold-PW | InvMSAFold-AR |
|---|---|---|---|
| Inter-Cluster | 15.8 | 4.6 | **0.49** |
| Intra-Cluster | 11.9 | 11.2 | **0.67** |

Table 1: Average KL Divergence across domains reported in Appendix A.3.2 between estimated densities of natural and synthetic samples projected on the first two PC components of the MSA for the Inter/Intra-Cluster datasets. For each method and for each backbone, we sampled 2k sequences. Densities are estimated through a Gaussian KDE of kernel size 1.0.

MSAs with synthetic samples from InvMSAFold-PW, InvMSAFold-AR and ESM-IF1, and compared the amino acid covariances in these MSAs with the covariances in the MSAs of natural sequences. Given an MSA of sampled or natural sequences, we define the covariance between amino acids $a$ and $b$ at positions $i$ and $j$ as $C_{ij}(a,b) = f_{ij}(a,b) - f_i(a)f_j(b)$, where $f_{ij}(a,b)$ is the frequency of finding amino acids $a$ and $b$ at these positions in the same sequence in the MSA and $f_i(a)$ and $f_j(b)$ are the overall frequencies of the amino acids $a$ and $b$ at these positions. In order to compare two sets of covariances, we calculate their Pearson correlation as in previous research (Trinquier et al., 2021). For details on these experiments, we refer to the Appendix A.3.1.

From Figure 5 and the attached table, we can see that InvMSAFold-AR and InvMSAFold-PW capture the covariances significantly better than ESM-IF1, especially at longer sequence lengths. In fact, InvMSAFold-AR shows the best performance overall, with a performance robust to $L$. The worse performance of ESM-IF1 in this task is not surprising, as it has not been trained for it, yet it signals a significant limitation of the model for many tasks. On the other hand, the stronger performance of InvMSAFold-AR compared to InvMSAFold-PW could be due to the latter being trained with pseudo-loglikelihoods, which do not capture covariances perfectly even when training a pairwise model directly (Ekeberg et al., 2013), or due to inefficient MCMC sampling. We also tested synthetic MSAs from ProteinMPNN (Dauparas et al., 2022), detecting an analogous behavior to ESM-IF1. We report the results in Appendix A.1.

The result strengthens our confidence that InvMSAFold is able to model the sequence landscape of an unseen structure. We explore this in more detail in the next section.

## 4.4 SEQUENCE PATTERNS

In this Section, we compare the projections of sampled sequences onto the first two PCA components of the one-hot encoded MSA of natural homologs to get a more fine-grained analysis of how the models are exploring the space of natural homologs compared to the global measure reported in Figure 5. Results for the NDP Kinase 1xqi (Pédelacq et al., 2005) are shown in Figure 6.The most striking feature is the narrow focus in this space for sequences generated from ESM-IF1, suggesting that the distribution of ESM-IF1 is too centered around a single sequence, with very small coverage of the full sequence space. Sequences sampled from InvMSAFold-PW show a broader coverage of

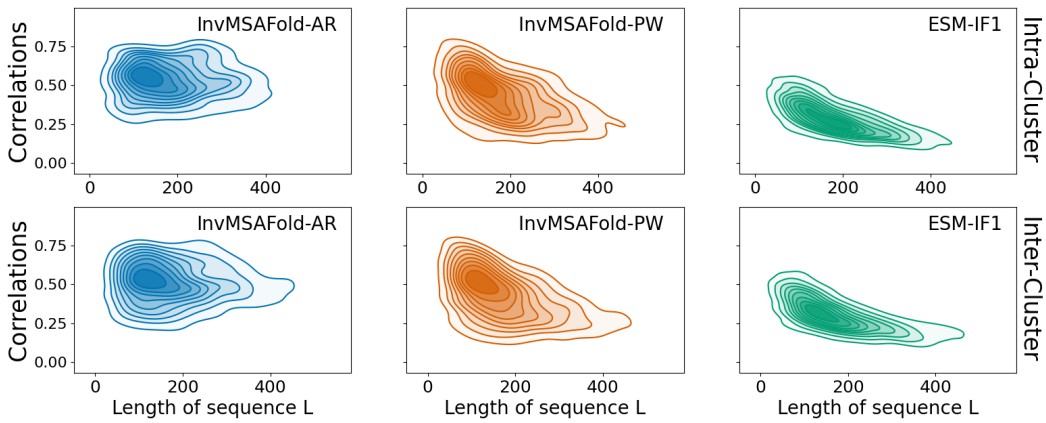

| **Quartiles** (↑) | **ESM-IF1** | | **InvMSAFold-PW** | | **InvMSAFold-AR** | |
|---|---|---|---|---|---|---|
| | Inter-Cluster | Intra-Cluster | Inter-Cluster | Intra-Cluster | Inter-Cluster | Intra-Cluster |
| First quartile | 0.23 | 0.21 | 0.29 | 0.28 | **0.37** | **0.35** |
| Median | 0.31 | 0.31 | 0.43 | 0.42 | **0.53** | **0.53** |
| Third quartile | 0.43 | 0.45 | 0.50 | 0.53 | **0.60** | **0.60** |

Figure 5: Distribution of Pearson's correlation coefficients between the covariances from sampled and natural sequences for domains having at least $2k$ natural sequences. **Top**: KDE plots representing how the correlations for InvMSAFold-AR (blue), InvMSAFold-PW (red) and ESM-IF1 (green) vary with sequence length $L$. The top/bottom rows show results for the Intra/Inter-Cluster test sets respectively. **Bottom**: Table with quartiles of the above Pearson's correlation coefficients. The best model for each quartile is highlighted by boldface numbers.

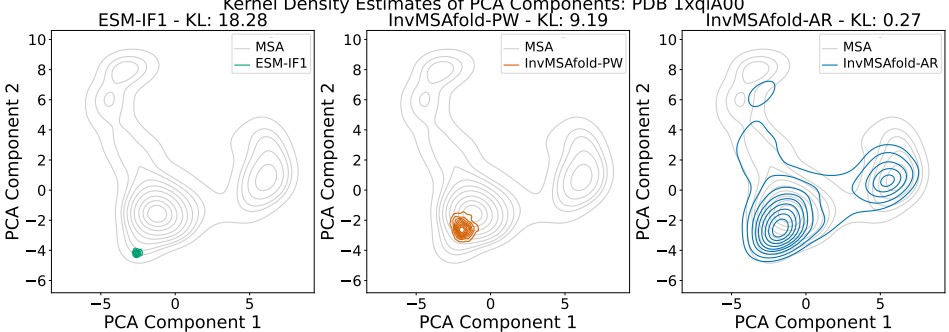

Figure 6: Estimated densities of natural and synthetic samples projected on the first two PC components of the MSA for 1xqiA00, with KL-divergence reported in the title of each subplot.

sequence space but are still concentrated around the single dominant mode. This again could be a result of the approximate training given the *pseudo likelihood*, or of the challenges of sampling from multimodal distributions for MCMC algorithms. While this increased diversity in the samples might already be useful in some settings, it is notable that in this case secondary modes are not captured at all, and that even the dominant mode is still covered only in part. In contrast, sequences sampled from InvMSAFold-AR cover the sequence space more comprehensively, demonstrating an ability to cover multiple distinct modes. These results are not restricted to this specific NDP Kinase, but are consistent across a variety of out-of-sample proteins we tested, as can be seen from Table 1. Other plots are reported in Appendix B.2, where we report also results for ProteinMPNN, which, although typically samples less narrowly, generally displays a similar performance to ESM-IF1.

## 4.5 PREDICTED STRUCTURES OF SAMPLED SEQUENCES

Similarly to other works (Li et al., 2023; Dauparas et al., 2022), we test to what extent the sequences generated by our models are predicted to fold into the correct structure as we increase the normalised hamming distance to the native one. From Figure 7 we can see that InvMSAFold-AR and InvMSAFold-PW are naturally capable of generating sequences with low sequence similarity to the native sequence, while ESM-IF1 often generates sequences very close to the native sequence, which is consistent with what we observed in Subsection 4.4.

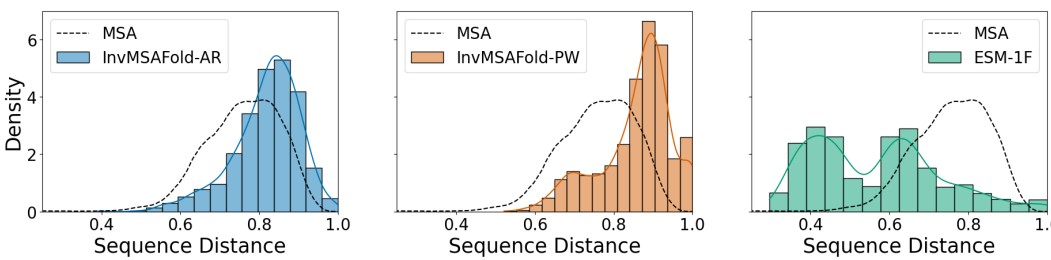

Figure 7: Distances to native sequences, averaged over 15 structures from the inter-cluster test set.

We therefore sampled sequences from ESM-IF1 at a higher temperature (for details of the sampling procedure we refer to the Appendix A.3.2). Following the generation of these sequences, we predicted their structures with AlphaFold 2 (Jumper et al., 2021; Mirdita et al., 2022) with templates switched off. From Figure 8, which shows results for the intra-cluster test set, we can see that for lower distances ESM-IF1 is the best-performing method. However, its performance drops significantly for larger distances, eventually becoming worse than both InvMSAFold-AR and InvMSAFold-PW, which show a more robust performance at higher distances. For the inter-cluster test set, we observe a similar behavior as the distance increases, with a general worsening of all of them (results reported in Appendix B.1). For the complete list of domains used from the two test sets in this experiment, we refer to Appendix A.3.2

## 4.6 PROTEIN PROPERTY SAMPLING

Another potential area of application for InvMSAFold are protein design tasks where other structural properties of the designed protein are important. In a virtual screening-like setting, one might want to generate a large number of sequences that obey certain structural constraints and then select from this set sequences that have other desirable properties. We test this setting by analyzing the range of (predicted) thermal stabilities and solubility of the sequences generated by the different models. In order to predict thermal stabilities we use Thermoprot (Erickson et al., 2022), which takes into account whether the protein operates within a Mesophile or Thermophile species. For solubility, we use predictions from Protein-Sol (Hebditch et al., 2017). Working with aligned sequence also introduces gaps, creating therefore variable effective length of the proteins (in terms of amino acids). We verified that the number of gaps was not discriminative for the two predicted properties. From Figure 9, which shows domain 1ny1A00, we observe that the support of InvMSAfold is significantly larger than of ESM-IF1, showing that the larger sequence diversity generated by both InvMSAFold-PW and InvMSAFold-AR translates into a wider range of predicted protein properties. Note also that

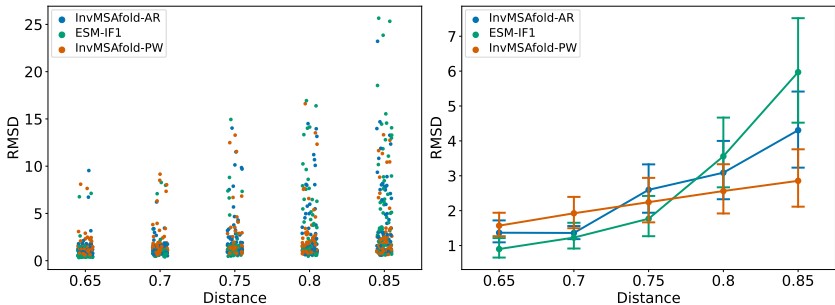

Figure 8: Comparison of the quality of the generated sequences at increasing hamming distance from native one. Shown is an average of 14 structures from the intra-cluster test set. We then refold the sequence with Alphafold and compare the refolded structure with the original one using RMSD.

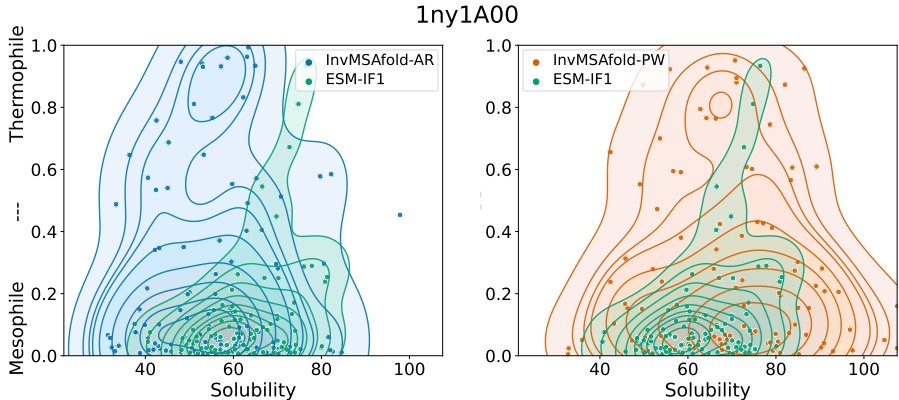

Figure 9: Comparison of the distribution of predicted solubility and thermostability of samples from InvMSAFold and ESM-IF1 for 1ny1A00. The x-axis reports the solubility predicted by Protein-Sol, while the y-axis the output of Thermoprot, a binary classifier (0 is mesophile and 1 thermophile).

while we generated the same number of sequences for a fair comparison, the computational resources are very different as we have shown in Subsection 4.2. These results are not specific to this domain, for more examples and for more details on the procedure we refer the reader to Appendix B.3.

## 5 CONCLUSION

InvMSAFold is a novel approach to inverse folding that combines modeling the complete sequence space corresponding to a structure with computational efficiency. The core idea is to frame the problem as a two-stage generation process: an expressive large neural network generates the parameters of a sufficiently expressive family-specific model. During training, the loss is computed on data augmented through an MSA, which captures the variability of sequences compatible with the input and prevents the model from narrowly focusing on the native sequence. Notably, this idea is not specific to our model formulation and could also be applied to other architectures, such as ESM-IF1. Once trained, for any input structure, the model requires only a single forward pass to generate the parameters of the pairwise model, which can then be used to generate diverse sequences orders of magnitude faster on CPUs compared to previous architectures. We extensively validate InvMSAFold through numerous out-of-sample tests, demonstrating its ability to generate sequences that deviate significantly from the native sequence while maintaining structural fidelity and capturing the evolutionary patterns of the MSAs. Additionally, we show that the broader exploration of the sequence domain enables the sampling of a wider range of protein properties of interest. We believe that InvMSAFold, with its combination of diverse sequence exploration and rapid sequence generation, opens up new possibilities for virtual screening, for example enabling simultaneous optimization of properties like thermostability, altered substrate specificity, and reduced toxicity.

## 6 ETHICAL CONSIDERATIONS AND CONCERNS

We recognize the importance of biosecurity considerations in protein design and the role of sequence-based screening systems. In principle, our approach generates diverse protein sequences that fold into known structures, which raises the question of how dangerous sequences are detected by existing screening methods.

To mitigate this risk, we emphasize that our method relies on homologous sequence data during training. This inherently means that any generated sequences remain within the evolutionary space of known proteins and would be detectable by homolog detection algorithms. Given that biosecurity screening protocols, such as those employed by DNA synthesis providers `https://genesynthesisconsortium.org/`, increasingly incorporate homology-based detection, we recommend that safeguards extend beyond simple sequence matching to include homologous sequence analysis for known hazardous proteins.

Moreover, existing tools for structure prediction and homology detection have reached a level of accuracy where they can reliably identify structurally and functionally similar sequences. The maturity of these technologies enables the detection of potentially dangerous sequences even when sequence similarity is low.

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

# A    APPENDIX

## A.1    COVARIANCE RECONSTRUCTION FOR PROTEINMPNN

In this Appendix, we report the results for another popular method available in the literature, Protein-MPNN (Dauparas et al., 2022). The architecture is built upon a transformer Vaswani et al. (2017) encoder-decoder architecture, similar to ESM-IF1 Hsu et al. (2022), but it is shallower as it consists of only three encoder and decoder layers. Moreover, the training is performed only on structures of the CATH dataset in contradistinction to ESM-IF1, which augments the training data with millions of structures predicted from Alpha-Fold (Jumper et al., 2021).

We begin with the covariance reconstruction capabilities of synthetic sequences generated from Protein-MPNN. The sampling procedure is analogous to that reported in Appendix A.3.1. To have consistency with the other models explored in Section 4.3, we sampled sequences at the training temperature of 1. From Figure 10, we can observe that the results are similar to those observed for ESM-IF1, as the recovered covariances are worse than both InvMSAFold-PW and InvMSAFold-AR, and the performance worsens significantly with sequence length. One difference with ESM-IF1 can be observed in the top-right plot of Figure 10; Protein-MPNN at temperature 1 can generate sequences with higher diversity, although still smaller than the one observed in the native MSAs. This could be due to the shallower architecture and the reduced training set of Protein-MPNN.

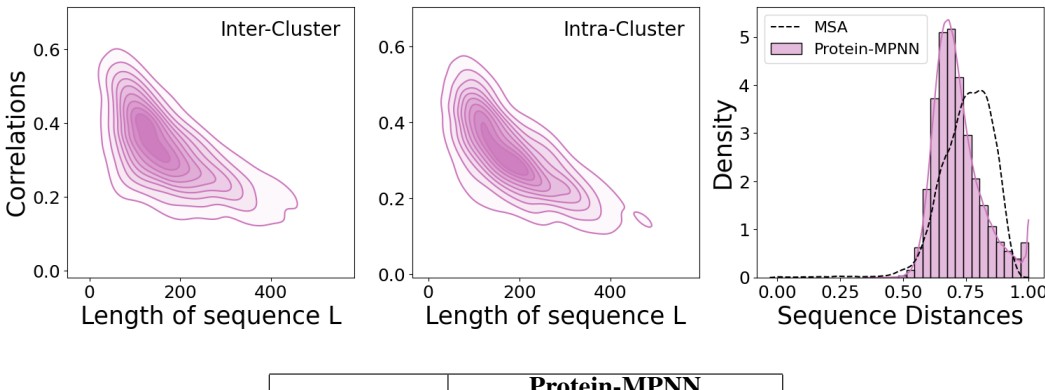

|  | **Protein-MPNN** | |
|---|---|---|
| **Quartiles** ($\uparrow$) | Inter-Cluster | Intra-Cluster |
| First quartile | 0.23 | 0.21 |
| Median | 0.31 | 0.31 |
| Third quartile | 0.43 | 0.45 |

Figure 10: **Top**: KDE plots representing how the correlations between covariances of sequences from Protein-MPNN and natural homologous vary with sequence length $L$ for sequences in the inter-cluster (left) and intra-cluster (center) test sets, and distances to native sequences for samples from Protein-MPNN, averaged over 15 structures from the inter-cluster test set (right). **Bottom**: Table with quartiles of the above Pearson's correlation coefficients.

## A.2 Training details

### A.2.1 Training and test curves

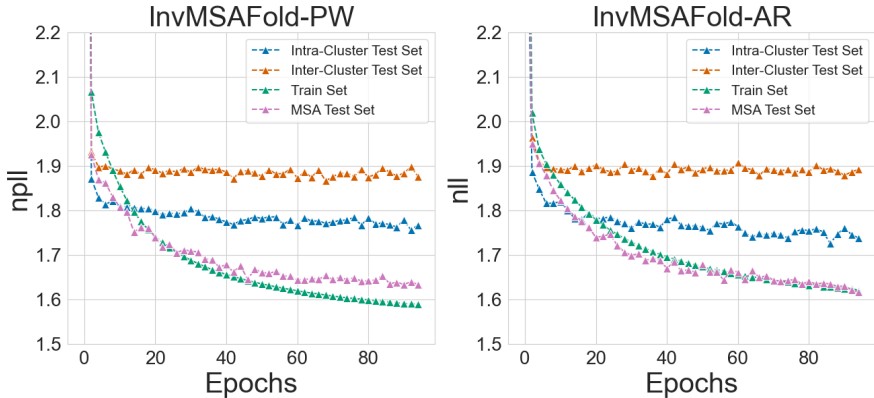

Figure 11: Negative pseudo log-likelihood (npll) for the train and test sets by training epochs. The training loss is a rolling average over a window of $5$ epochs, which is why the plotted value is higher than the test losses initially.

As can be seen from Fig. 11, for both models the ordering of the losses on the different datasets is consistent with the reasoning behind the split in the last section: The *MSA* test set has sequences coming from domains which the model has seen during training, and the loss on this test set is similar to the training loss. The *intra-cluster* test set contains domains of which the model has seen domains belonging to the same sequence cluster; the loss on this test set is higher than the training loss and seems to saturate during training. For the *inter-cluster* test dataset, which contains domains that belong to sequence clusters not present in the training set, the loss is significantly higher, even though still much better than random.

### A.2.2 Hypertuning

To select hyperparameters for InvMSAFold-AR we relied on the library Optuna. The parameters we optimized for where Adam's learning rate($lr$), the rank of the approximation for the coupling matrix $J(K)$, the penalties for the couplings and the fields $(\lambda_J, \lambda_H)$, the batch size used for training $(B)$, the batch MSA size used in the loss $(|M_X|)$ and the value of dropout. We sampled parameters through the TPESampler, allowing median pruning of bad trials to improve the speed. We gave Optuna $50$ trials of 90 epochs each (irrespective of batch size), while as a selecting metric, we used the average of the log-likelihood between the inter/intra-cluster test dataset. In the table below we recap the parameter values selected by the hyper-tuning

| Hypertuning results | | | | | | |
|---|---|---|---|---|---|---|
| Model | Dropout | B | M | K | $(\lambda_J, \lambda_h)$ | lr |
| ArDCA | 0.1 | 8 | 32 | 48 | (3.2e-6, 5.0e-5) | 3.4e-4 |

Table 2: Parameters selected arDCA by hyperparamter optimization

We applied an identical strategy for InvMSAFold-PW, yet we found that in many applications the tuned model slightly underperformed the original one we were previously using whose hyperparameter values are the ones reported in Section 2.

## A.3 EXPERIMENTS DETAILS

### A.3.1 SECOND ORDER RECONSTRUCTION

The experimental procedure for both the inter and intra cluster datasets can be organized in the following steps:

1. We filtered the structures, keeping only those for which the MSA generated by MMseqs2 has at least $2k$ sequences. This is because small MSAs tend to give very noisy covariances which could bias the benchmarking.

2. For every sequence in the filtered dataset, we generate $10k$ synthetic samples for all three models. To generate the samples from InvMSAFold-PW we use the library bmDCA, running 10 parallel chains of standard Metropolis-Hastings MCMC algorithm and pooling the results, for ESM-IF1 we used the built-in sampler while for InvMSAFold-AR we built our own sampler.

3. For ESM-IF1, once generated the samples, we re-aligned them using the full MSA to get a fair comparison. Note that, while InvMSAFold-PW and InvMSAFold-AR have seen gaps during training, ESM-IF1 was trained on the gapless native sequence. It therefore produces un-aligned sequences. We used the PyHMMER library for alignment tasks.

4. Given the samples, we compute the covariance matrices of the generated samples and of the true MSA. We then compute the Pearson correlation between the flattened versions.

### A.3.2 PREDICTED STRUCTURES OF SAMPLED SEQUENCES

First, we report the list of the domains used from the intra/inter-cluster test set in this experiment

- **intra-cluster**: 5i0qB02, 4iulB00, 2wfhA00, 2egcA00, 3oz6B02, 3m8bA01, 2pz0B00, 2cnxA00, 4k7zA02, 1dl2A00, 1vjkA00, 1udkA00, 3bs9A00.

- **inter-cluster**: 1otjA00, 2ytyA00, 1p9hA00, 2o0bA02, 1b34B00, 1f6mA02, 2hs5A01, 2bh8A02, 2de6A03, 1ia6A00, 4gc1A01, 3sobB02, 1xqiA00, 4yt9A01.

Synthetic sequences from both InvMSAFold-AR and InvMSAFold-PW produce very diverse sequences without further intervention. On the other hand, ESM-IF1 produces sequences whose hamming distance from the native sequence is significantly lower. Given that we want to test the different models' ability to recover the native structure from sequences having high dissimilarity from the native one, we leverage the built-in temperature parameter of the model to get such sequences from ESM-IF1, increasing it to sample more diverse sequences.

Specifically, InvMSAFold-PW and InvMSAFold-AR generate sequences having normalized hamming distance between $0.65$ and $0.9$. We hence split the interval $[0.65, 0.9]$ into 5 bins of equal width of $0.05$.

We use the following procedure to fill each of these bins with at least 10 sequences from ESM-IF1:

1. Generate 1000 samples from ESM-IF1.

2. Fill the bins that still are lacking some sequences.

3. If all the bins are full, return the binned sequences, otherwise increase the temperature of ESM-IF1 by $0.1$ and go back to step 1.

For InvMSAFold-PW and InvMSAFold-AR we just generate $10k$ sequences each for each domain, and then fill the bins relative to every domain.

# B ADDITIONAL PLOTS

## B.1 PREDICTED STRUCTURES FOR INTER-CLUSTER TEST SET

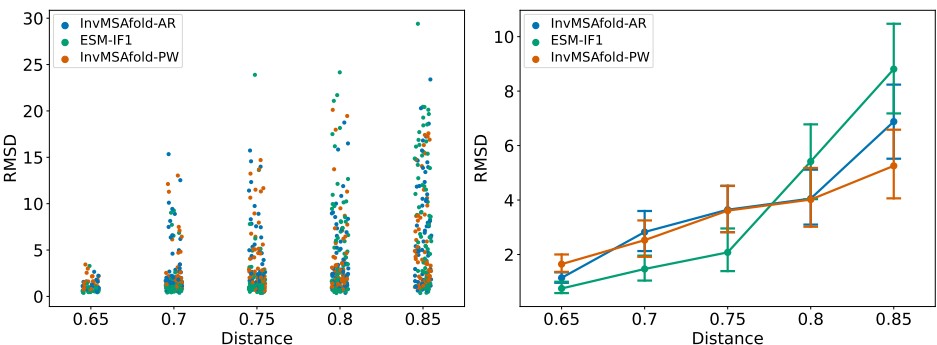

Figure 12: Comparison of the quality of the generated sequences at increasing hamming distance from native one. Shown is an average of 14 structures from the inter-cluster test set. We then refold the sequence with Alphafold and compare the refolded structure with the original one using RMSD.

## B.2 PCA with also ProteinMPNN

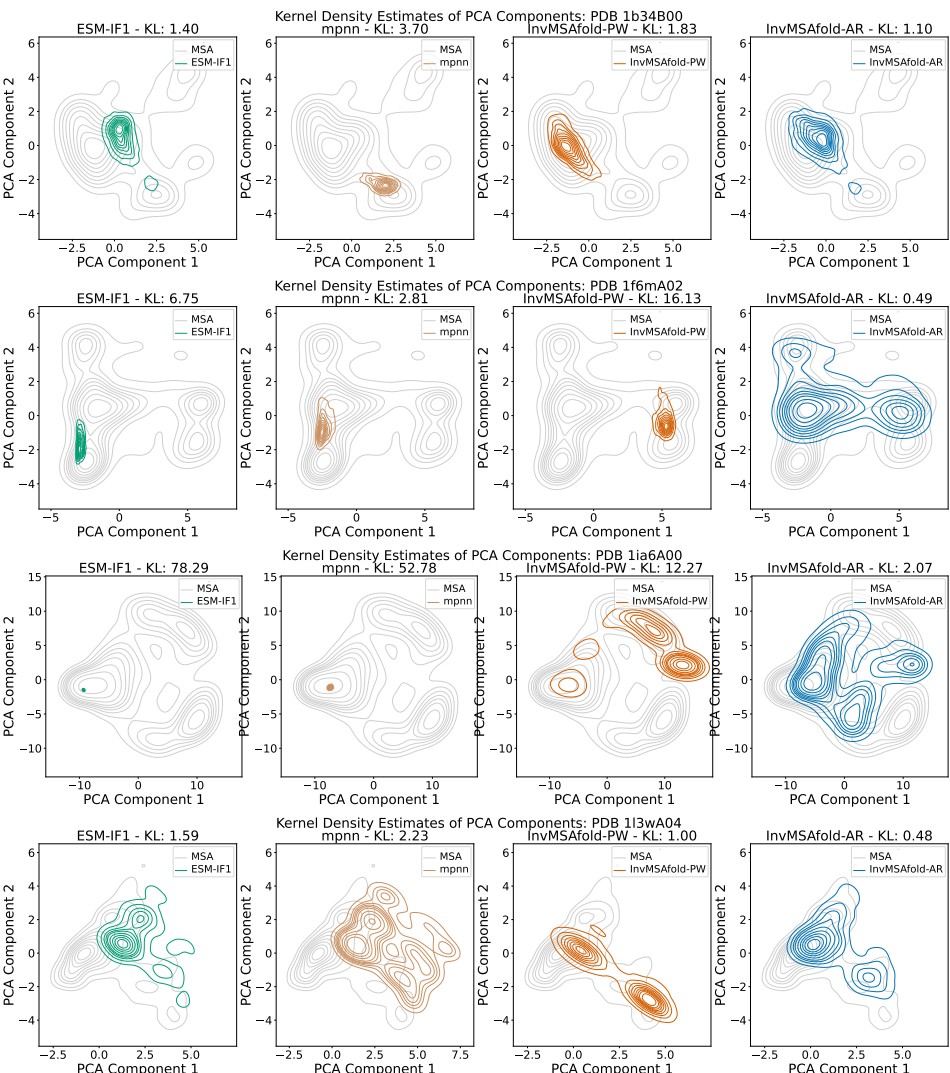

Figure 13: Sampled sequences projected onto the first two PCA components of natural sequences for various PDBs. We also used this density estimate to compute the Kullbach-Leiber (KL) divergence between the density of the natural data and the density of the sampled data. The values of these KLs are written in the title of each subplots.

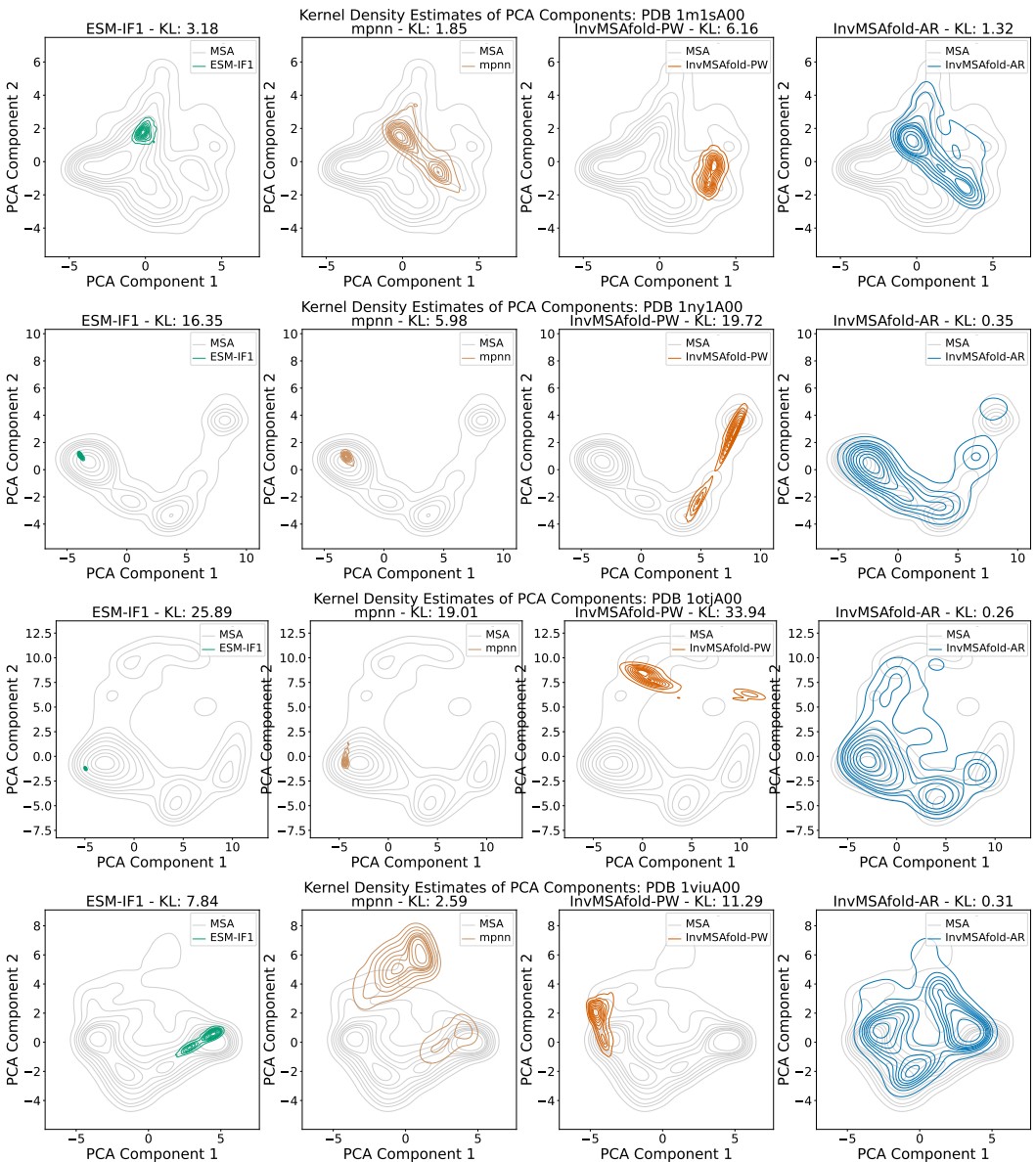

Figure 14: Sampled sequences projected onto the first two PCA components of natural sequences for various PDBs. We also used this density estimate to compute the Kullbach-Leiber (KL) divergence between the density of the natural data and the density of the sampled data. The values of these KLs are written in the title of each subplots.

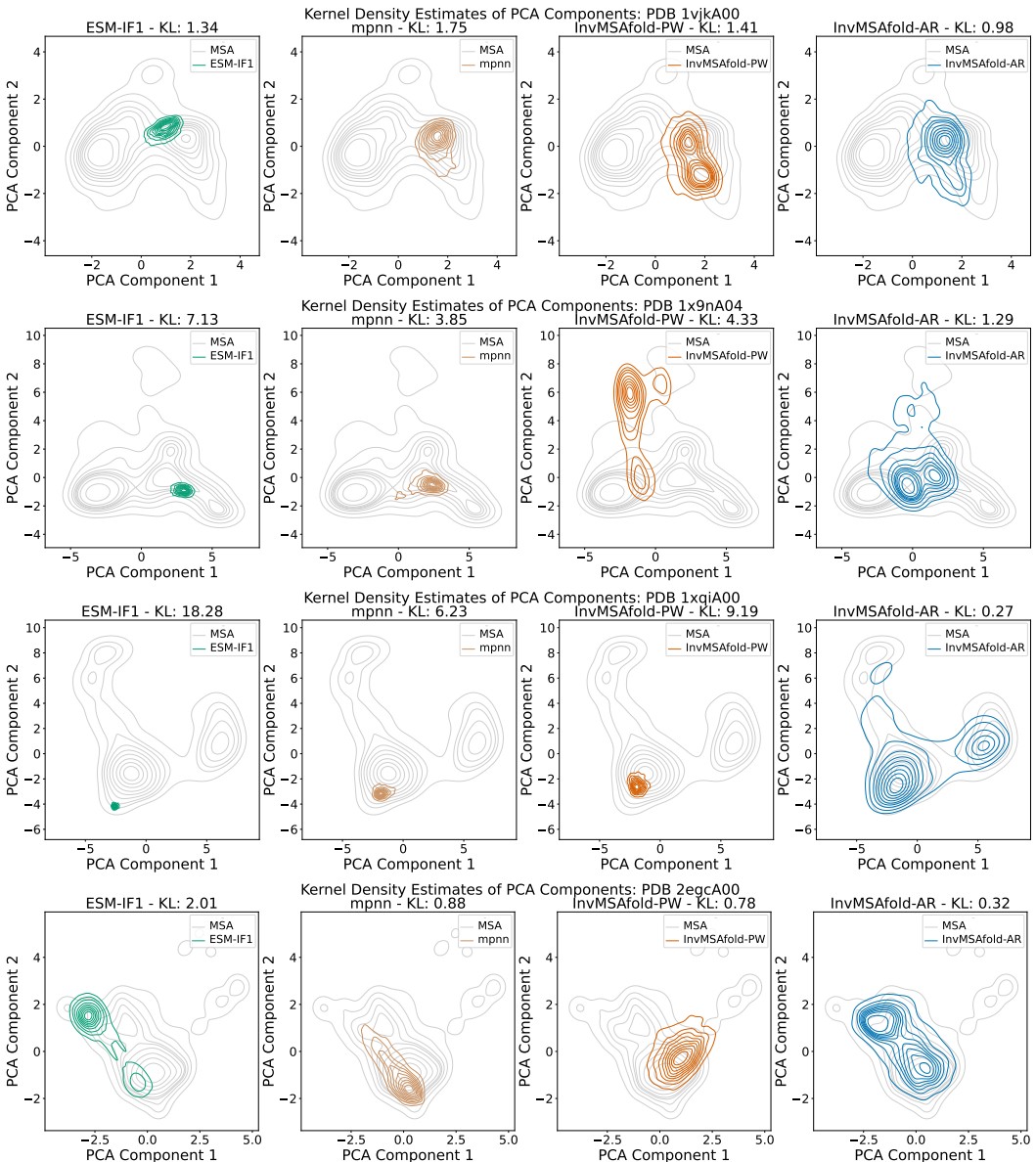

Figure 15: Sampled sequences projected onto the first two PCA components of natural sequences for various PDBs. We also used this density estimate to compute the Kullbach-Leiber (KL) divergence between the density of the natural data and the density of the sampled data. The values of these KLs are written in the title of each subplots.

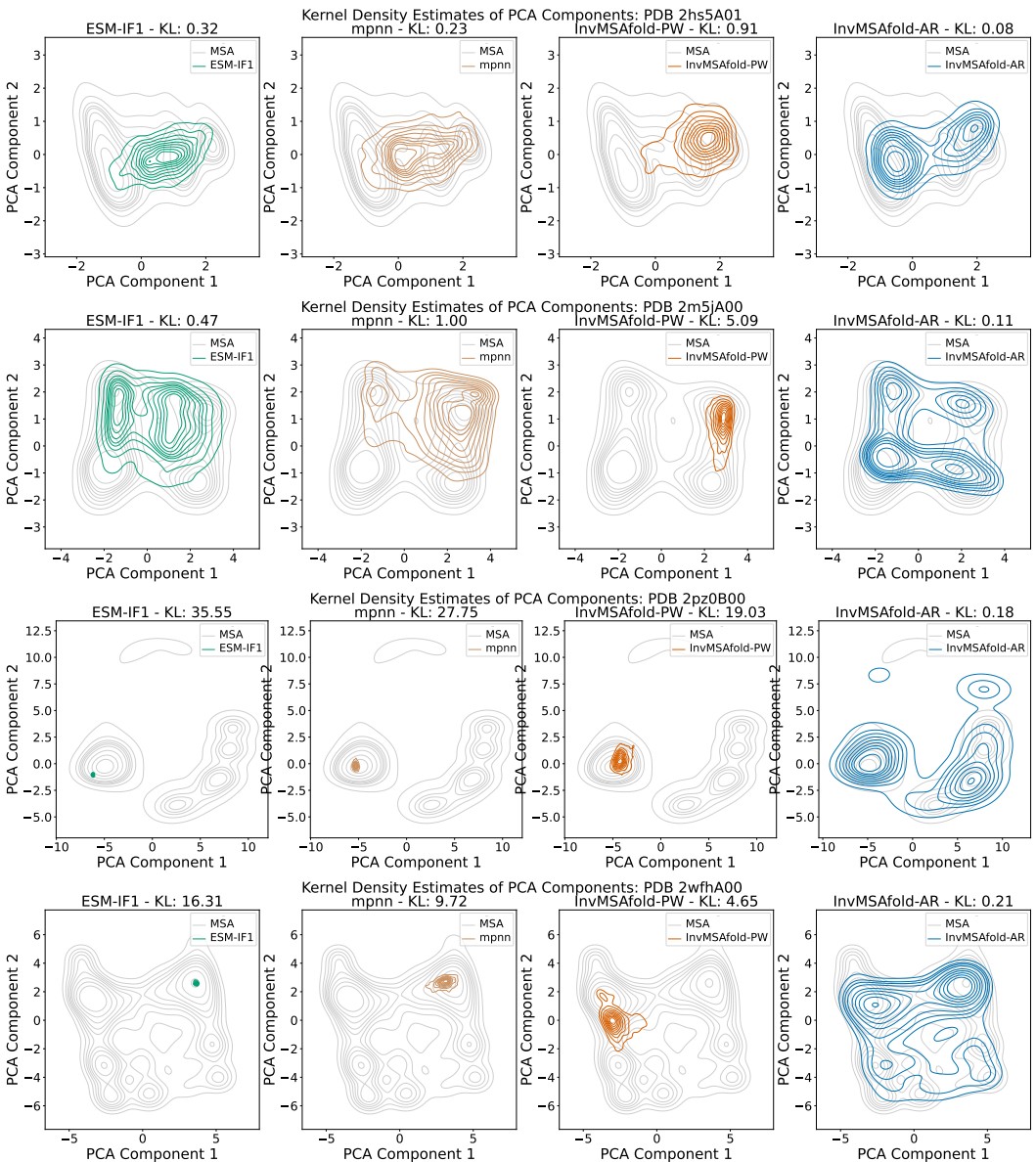

Figure 16: Sampled sequences projected onto the first two PCA components of natural sequences for various PDBs. We also used this density estimate to compute the Kullbach-Leiber (KL) divergence between the density of the natural data and the density of the sampled data. The values of these KLs are written in the title of each subplots.

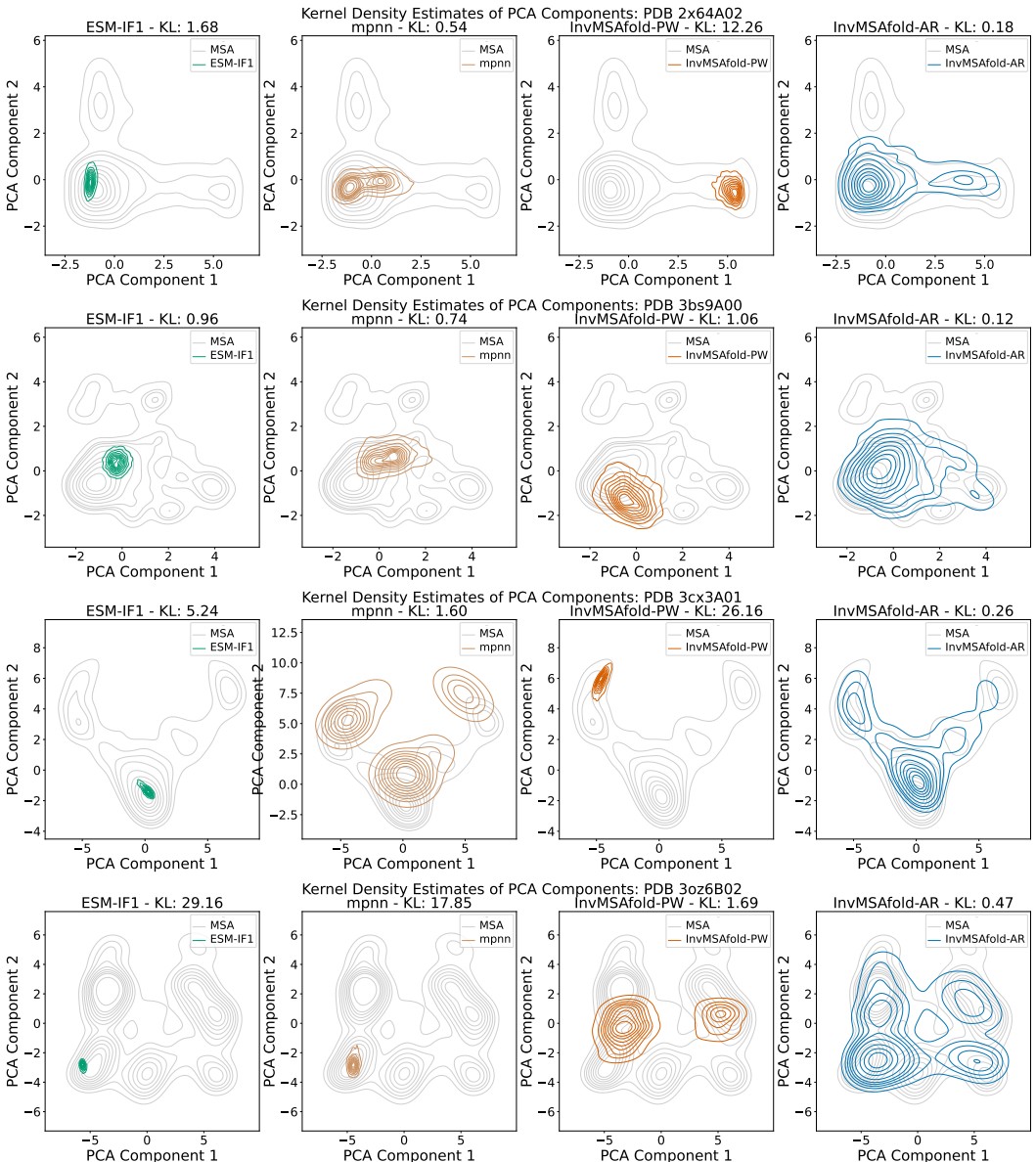

Figure 17: Sampled sequences projected onto the first two PCA components of natural sequences for various PDBs. We also used this density estimate to compute the Kullbach-Leiber (KL) divergence between the density of the natural data and the density of the sampled data. The values of these KLs are written in the title of each subplots.

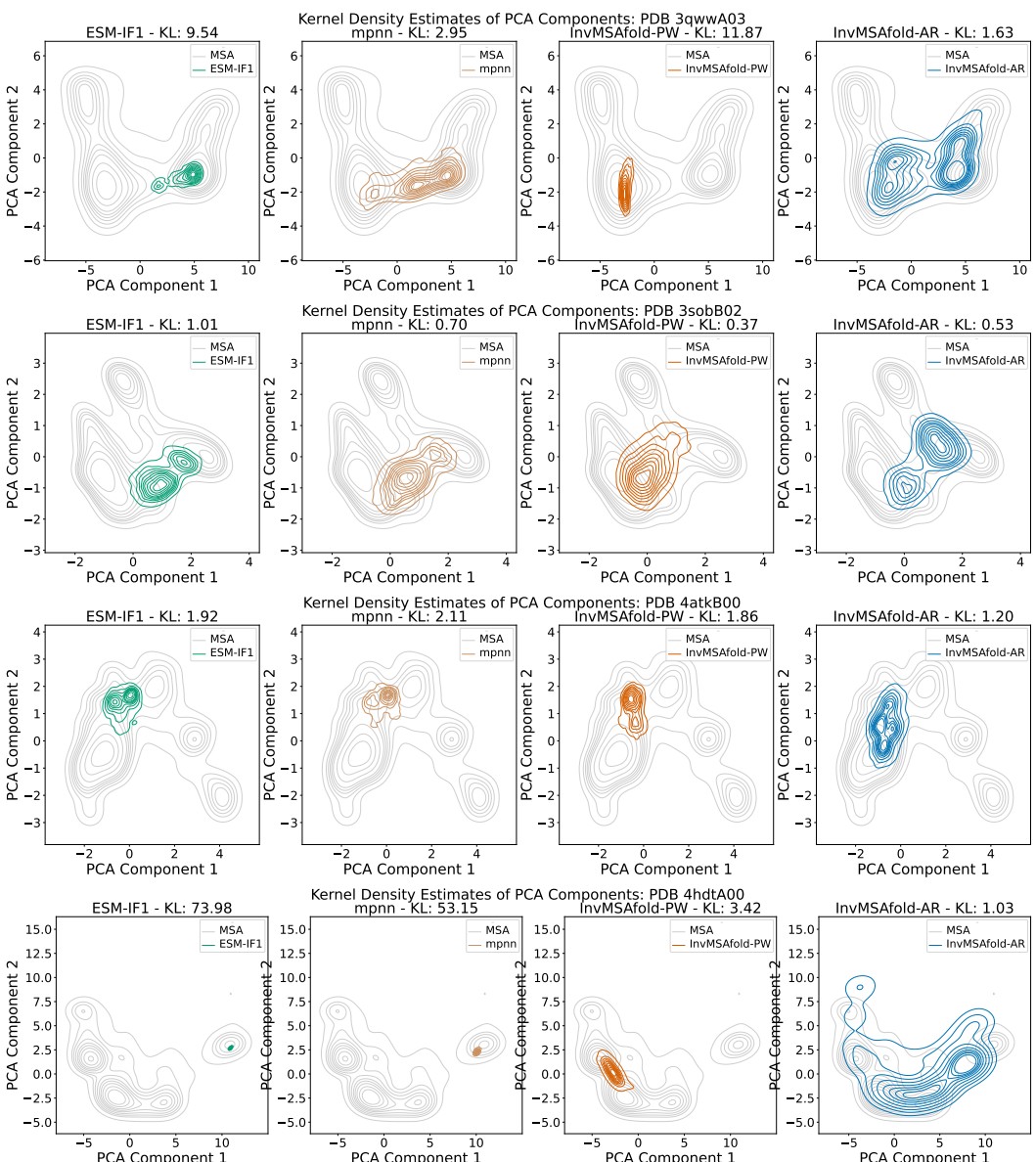

Figure 18: Sampled sequences projected onto the first two PCA components of natural sequences for various PDBs. We also used this density estimate to compute the Kullbach-Leiber (KL) divergence between the density of the natural data and the density of the sampled data. The values of these KLs are written in the title of each subplots.

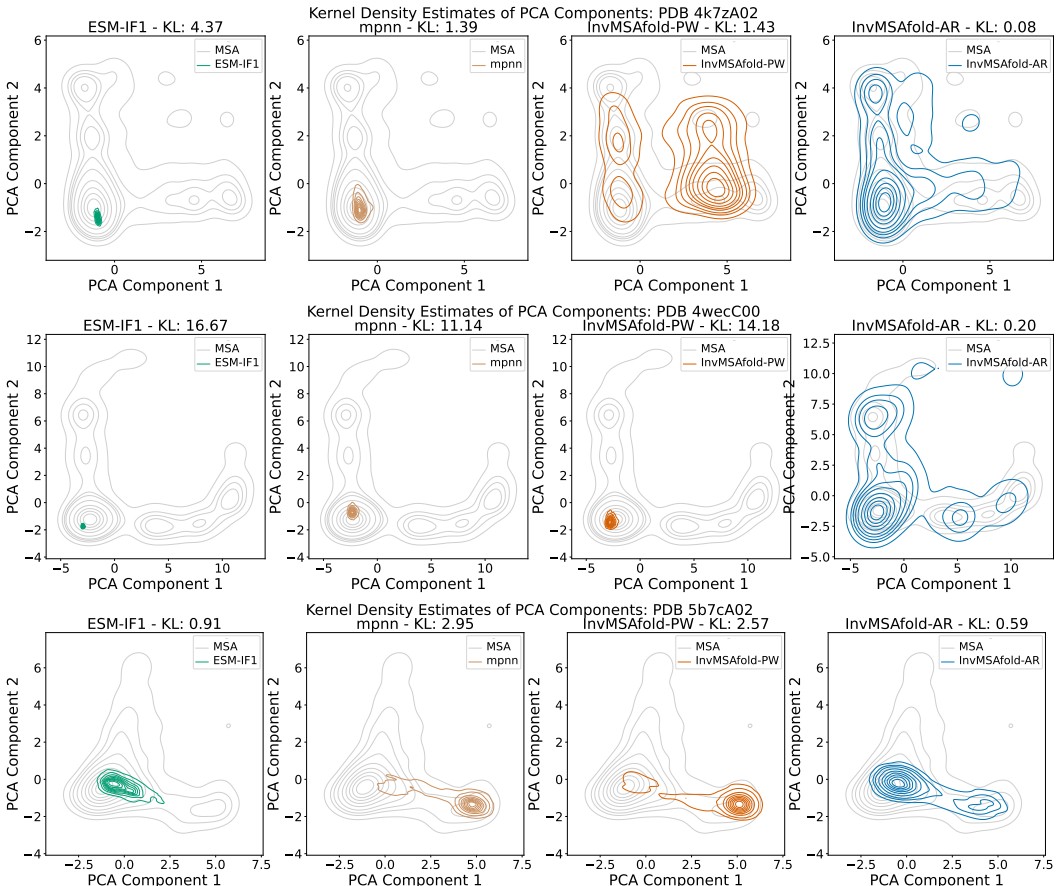

Figure 19: Sampled sequences projected onto the first two PCA components of natural sequences for various PDBs. We also used this density estimate to compute the Kullbach-Leiber (KL) divergence between the density of the natural data and the density of the sampled data. The values of these KLs are written in the title of each subplots.

## B.3    BIVARIATE PLOTS: THERMOSTABILITY VS SOLUBILITY

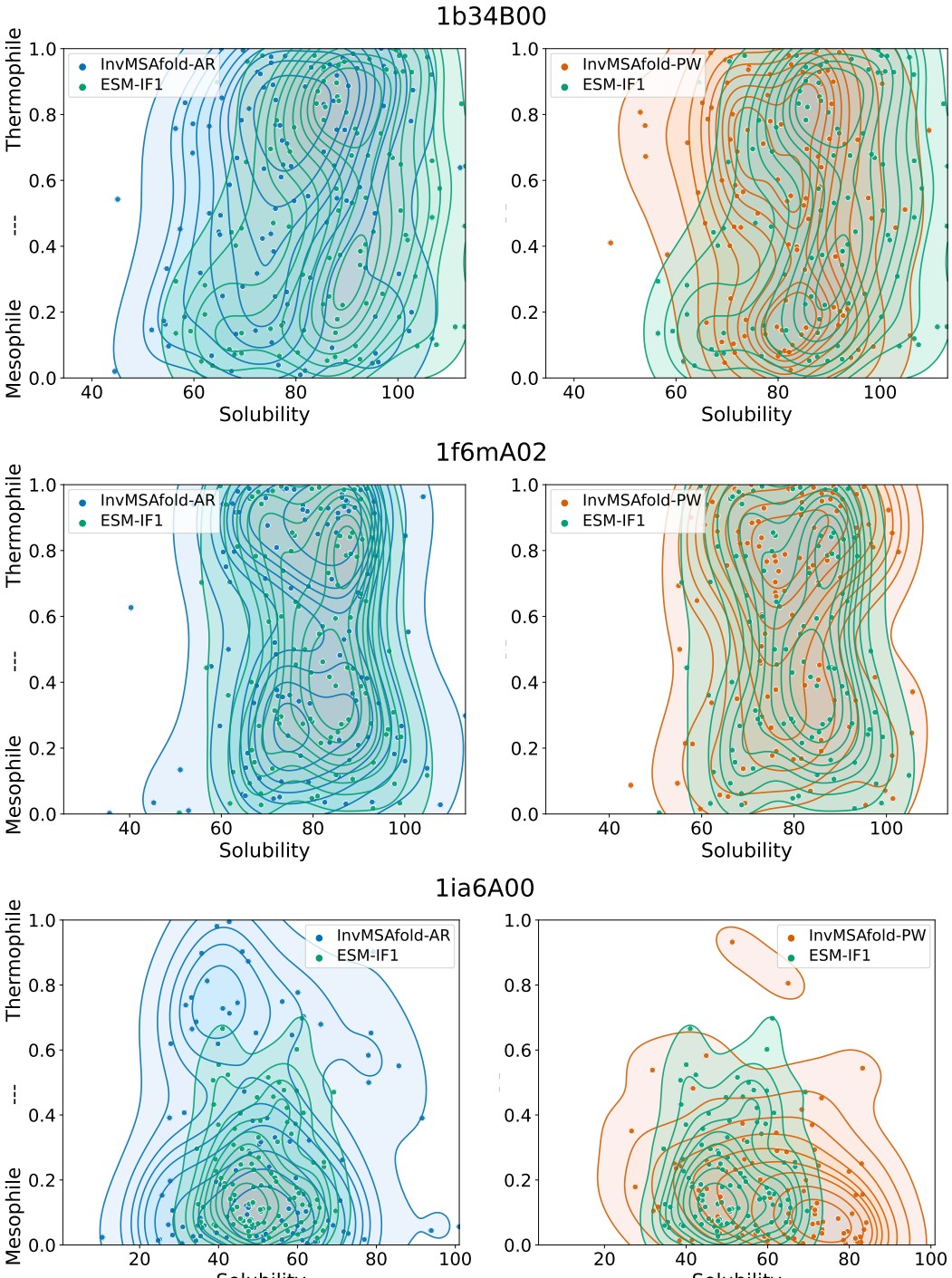

Figure 20: Comparison of the distribution of predicted solubility an thermostability of samples generated with InvMSAFold and ESM-IF1. These plots show the results for domain 1ia6A00, 1f6mA02, 1b34B00.

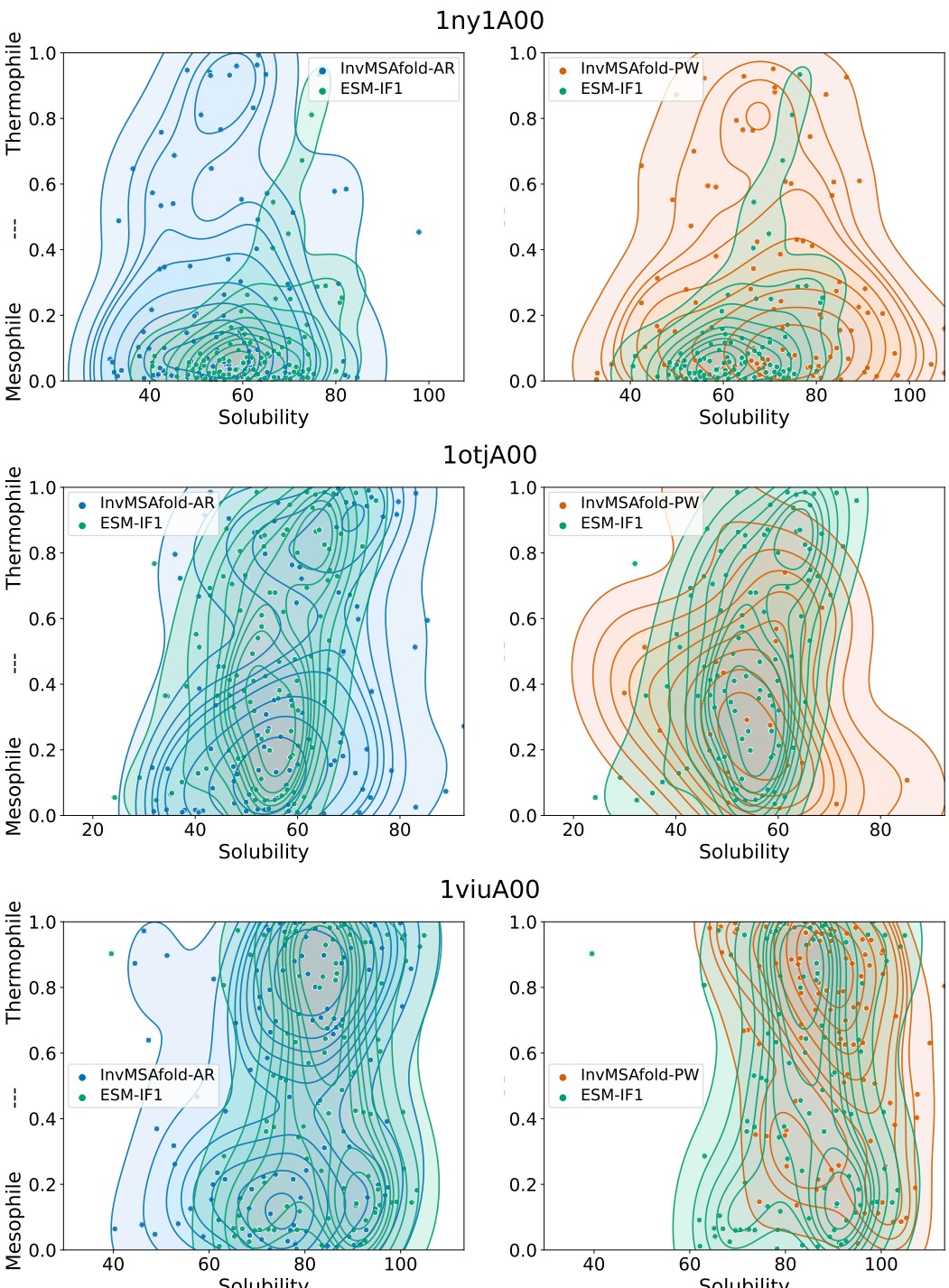

Figure 21: Comparison of the distribution of predicted solubility an thermostability of samples generated with InvMSAFold and ESM-IF1. These plots show the results for domain 1ny1A00, 1otjA00, 1viuA00.

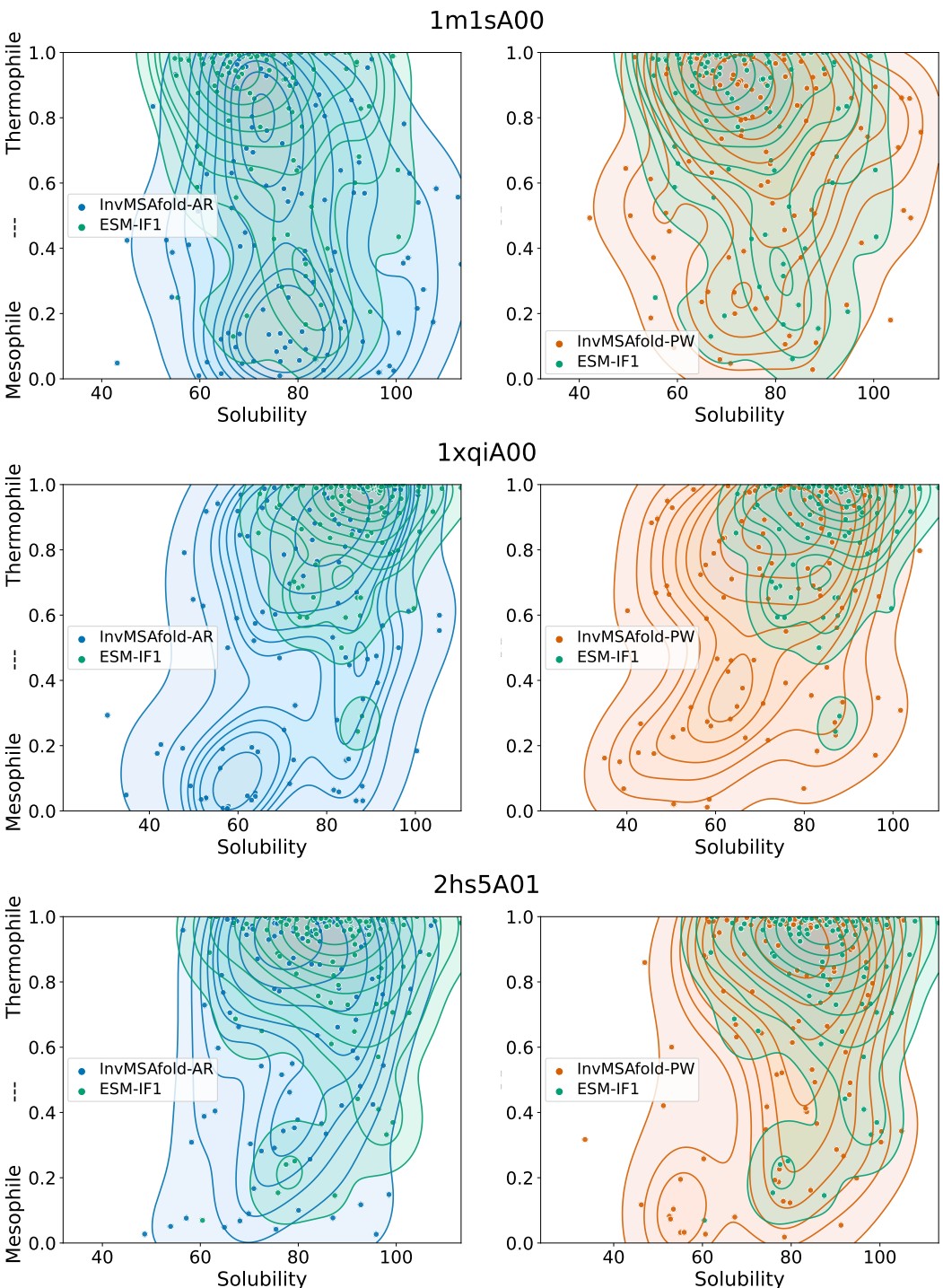

Figure 22: Comparison of the distribution of predicted solubility an thermostability of samples generated with InvMSAFold and ESM-IF1. These plots show the results for domain 1m1sA00, 1xqiA00, 2hs5A01.

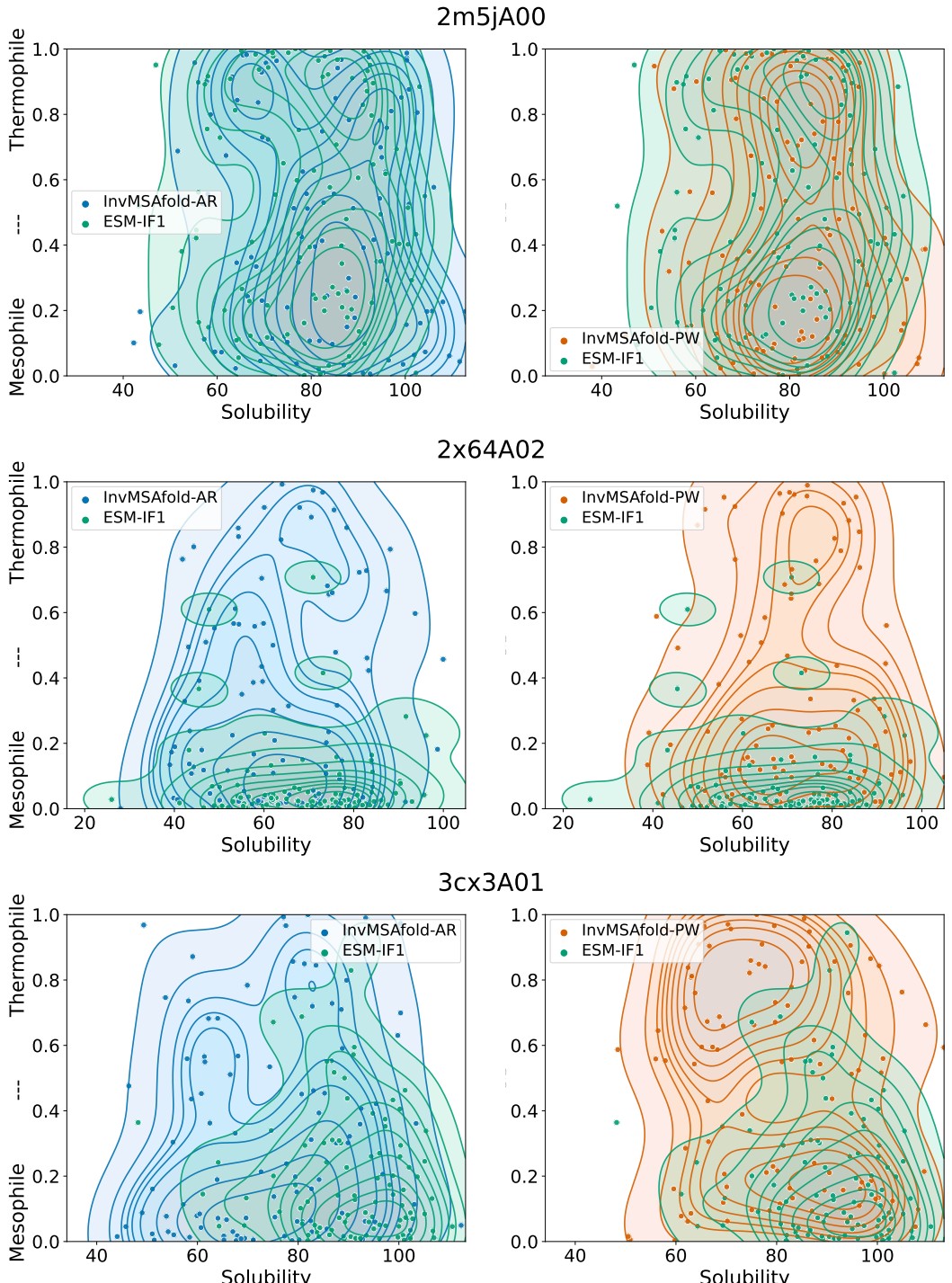

Figure 23: Comparison of the distribution of predicted solubility an thermostability of samples generated with InvMSAFold and ESM-IF1. These plots show the results for domain 2m5jA00, 2x64A02, 3cx3A01.

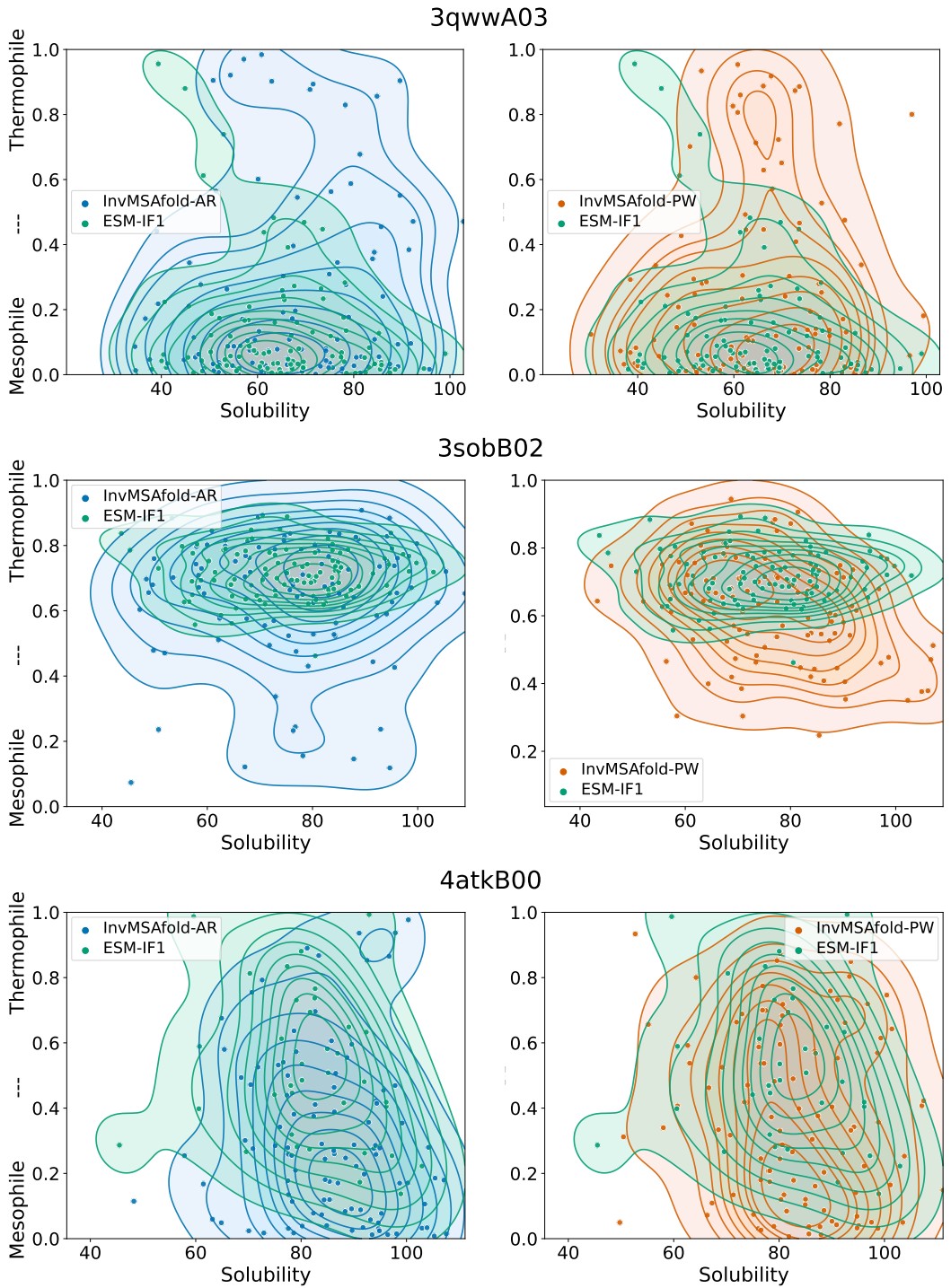

Figure 24: Comparison of the distribution of predicted solubility an thermostability of samples generated with InvMSAFold and ESM-IF1. These plots show the results for domain 3qwwA03, 3sobB02, 4atkB00.

