# OpenReview forum: "Fast Uncovering of Protein Sequence Diversity from Structure"
_ICLR.cc/2025/Conference — ICLR 2025 Spotlight_

### Official Review · Reviewer_D2uA · 2024-10-23

**Soundness:** 2
**Presentation:** 4
**Contribution:** 2
**Rating:** 8
**Confidence:** 4

**Summary:**

In this paper, the authors present an efficient method for designing protein backbones using a neural network that predicts a Potts model. The proposed architecture includes a pre-trained ESM1-IF encoder that encodes the protein backbone, generating rotation-invariant embeddings. These embeddings are then passed through a transformer-based decoder, which produces a low-rank matrix that is ultimately used to compute the fields and couplings. The low-rank approximation is a clever technique that helps mitigate the quadratic scaling cost typically associated with such computations. The neural network was trained using two distinct approaches: (1) a standard pseudo-likelihood loss, and (2) autoregressive sampling (over amino acids) with maximum likelihood training. To avoid training on single sequences, the model was trained on multiple sequence alignments (MSAs), with the mean pseudo-negative log-likelihood calculated over randomly sampled subsets of the MSA. Training and testing data were sourced from the CATH database, following its hierarchical classification to create test sets of varying difficulty, depending on the similarity between the training and test data. The authors demonstrate that their model better reconstructs covariance matrices compared to ESM1-IF, based on Pearson correlations. Moreover, the authors show that projected MSAs using PCA more closely reflect the natural sequence distributions, suggesting that their generated sequences, or predicted MSAs, are more diverse. When refolding designed sequences for test set structures, the InvMSAfold method proves to be more robust than ESM1-IF for sequences that deviate further from the native structure, and comparable to ESM1-IF for sequences that are highly similar to the native.
In conclusion, the paper demonstrates how a Potts model can be efficiently constructed, showing that the resulting model generates sequences that are plausible, diverse, refold successfully with AlphaFold2, and possess other promising biochemical attributes.

**Strengths:**

The authors present a novel and elegant approach to optimizing Potts model construction, training and sampling. The paper is well-structured, clearly outlining each crucial part of the methodology in a way that is easy to follow. The method is compared and benchmarked against a well-established approach, and performance metrics computed and reported.

**Weaknesses:**

While the proposed methodology for improving the efficiency of Potts model construction is promising, there are a few areas where the paper could be strengthened. First, Potts models have long been used in fixed-backbone protein design, which makes it difficult to clearly identify the novelty and specific contributions of this work. Additionally, the method relies on components of ESM1-IF and then benchmarks against this model, which may limit the fairness or objectivity of the comparison. Another area for improvement is scalability. The paper does not provide any analysis on how the model handles large structures or long sequences, which could be useful for evaluating its broader applicability. Furthermore, there is no discussion on the significance of using MSAs for training versus single-sequence training, nor is there any exploration of how deep the MSAs need to be if they are indeed important.

**Questions:**

I'd like to raise a few major points:
* It would strengthen the paper to benchmark against other methods as well. I would suggest for example a simple Potts model without the low-rank approximation and without the pre-trained ESM1-IF encoder, and an additional method such as ProteinMPNN beyond ESM1-IF. This would highlight the contributions of the paper more clearly, as currently, it may seem somewhat reliant on ESM1-IF.
* I highly recommend adding a plot that shows RMSD versus sequence recovery, as these metrics would provide valuable insights into the model’s performance.
* In Section 2.2.1, the explanation of Equation 7 and how it maintains linear scaling isn’t entirely obvious, at least to me. I suggest elaborating on this either within the main text or in the supplementary material to clarify the reasoning. It would be helpful to include 1-2 sentences explaining why the method or process is linear and how this linearity is established. This will provide clarity to the reader and strengthen the argument by highlighting the underlying reasoning behind the concept.
* To make the manuscript even stronger, it would be useful to include (1) an analysis of how the method scales with very large sequences or structures, and (2) a discussion of how the size of the MSA impacts model performance.

A minor point:
* In Section 2.2, I recommend including the formula that shows the normalization constant, as it is referenced in the text but not explicitly provided.

There are several typos throughout the manuscript that disrupt the flow. I have listed the ones I noticed while reading, but I recommend a re-read of the manuscript to specifically check for additional typos:
   - Line 17: The phrase “space of sequences with pairwise interwise interactions, capturing the amino acid…” contains the term “interwise,” which doesn’t seem correct or clear.
   - Lines 107-108: The word "Moreover" is used consecutively, which disrupts the flow.
   - Line 117: The word "Whos" should be corrected to "Whose."
   - Line 299: "We monitor the the negative..."—"the" is repeated.
   - Line 315: "A can be seen..." should likely be "As can be seen..."

---

> ### Author Response · Authors · 2024-11-23
>
> We first want to thank the reviewer for carefully reading the manuscript and for the questions raised, which is allowing us to improve the manuscript. We will address the questions point by point, reporting the original question in _italics_ followed by the answer.
>
> 1) _It would strengthen the paper to benchmark against other methods as well.:_ We thank the reviewer for raising this point, which has also been brought to our attention by another reviewer. To improve the manuscript we are hence also comparing InvMSAFold-AR/PW with [ProteinMPNN](https://www.science.org/doi/10.1126/science.add2187), which will then be added to the manuscript.
> 2) _I highly recommend adding a plot that shows RMSD versus sequence recovery, as these metrics would provide valuable insights into the model’s performance_:     We thank the reviewer for this comment, which allows us to clarify some of our experiments and improve the paper. Indeed we agree that assessing the relationship between sequence recovery and predicted RMSD of generated sequences is a key metric to evaluate the different models analyzed. As originally conjectured, and provided evidence for in Figure 7 of the paper, ESM-IF1 and our InvMSAFold architectures sample generally at different hamming distances from the native sequence; indeed the former focuses too narrowly on the native sequence, while both InvMSAFold-PW and InvMSAFold-AR explore sequences whose distance from the native is more consistent with those observed in true MSAs. Given this fact, to make more of an "apple to apple" comparison between sequences from the the different models we also needed samples from ESM-IF1 at higher hamming distances, which can be achieved by leveraging the built-in temperature parameter of ESM-IF1 sampler. This is precisely what we do in Section 4.5 and is reported in Figure 8. To highlight the dependence between sequence recovery and RMSD for the different models we decided to bin sequences based on their distance from the native. As can be seen from Figure 8, the RMSD of predicted structures from sequences generated from ESM-IF1 deteriorates much more rapidly compared to those from both InvMSAFold-PW and InvMSAFold-AR, which we feel addresses the comment raised by the reviewer. For more details on the experimental procedure we refer the reviewer to appendix A.2.2.
> 3) _In Section 2.2.1, the explanation of Equation 7 and how it maintains linear scaling isn’t entirely obvious, at least to me_: We thank the reviewer for raising this point, which allowed us to better clarify a crucial aspect of the architecture. The reason why Eq.(7) proves  linear scaling, is that on the left side of Eq.(7) we have a double sum over the amino acid positions $\sum_{i<j}$, resulting in a quadratic cost, while through careful computations on the right side of Eq.(7) we only have single sums over the position $\sum_i$, resulting in a linear cost. In the updated version of the manuscript we are making sure to make clear how linearity is established and why Eq.(7) achieves linearity.
> 4) _To make the manuscript even stronger, it would be useful to include (1) an analysis of how the method scales with very large sequences or structures, and (2) a discussion of how the size of the MSA impacts model performance.:_     We thank the reviewer for raising this point, which allows us to make the manuscript clearer. During training we capped the domains length to $512$ as this included almost all domains in the CATH dataset, and resulted in a significant reduction in memory requirements to run the model. In any case, the built-in attention layers of PyTorch are capped to a maximum input length of $1024$, hence we could not consider sequences with length higher than that. We interpret the second point of the reviewer as the size of the MSA used during training, as during testing the MSA is not needed to run the model. In this case, we agree with the reviewer that assessing the relationship between the MSA size and model performance is of great interest. Indeed, for this specific reason we included the MSA size as one of the parameters to be tuned in hypertuning reported in Appendix A.1.2. From there we observed that as soon as the MSA size is large enough, where this threshold heuristically seems to be around $32$, then the observe comparable performance for the different models trained with all other parameters equal. We have highlighted more this aspect in the revised version of the manuscript.
> 5) _In Section 2.2, I recommend including the formula that shows the normalization constant, as it is referenced in the text but not explicitly provided._ : We will add the normalization constant in the updated manuscript as suggested by the reviewer
> 6) We further thank the reviewer for reporting the typos/grammar above, which allowed us to polish the manuscript. We are re-reading the manuscript with great care and correcting all grammatical errors to make sure the updated manuscript will not have any.

---

> > ### Comment · Reviewer_D2uA · 2024-11-25
> >
> > I would like to thank the authors for clarifying how eq 7 ensures linear scaling—this is now clear to me. I also now understand the challenges associated with testing very large sequences and the related MSA cropping. That said, I was unable to find a comparison of the method to ProteinMPNN, which I believe is an important aspect to address. Could the authors kindly point me to the relevant table or consider updating the manuscript to include this comparison?

---

> > > ### Author Response · Authors · 2024-11-25
> > >
> > > We thank the reviewer for the feedback on our previous answers.
> > >
> > > We completely agree with the reviewer on the importance of comparing with ProteinMPNN, and we are running some comparison experiments at the time of this message. We just wanted to do a single revision of the manuscript which addressed all the questions raised by the reviewers at once, while also inserting their suggestion within the flow of the manuscript.
> > > We are hence just finishing this process, and we plan to upload an updated version of the manuscript tomorrow, were the reviewer will be able to find the results on ProteinMPNN.

---

> > > > ### Comment · Reviewer_D2uA · 2024-11-27
> > > >
> > > > I thank the authors for addressing my concerns. In light of their response, I have revised my rating to 8.

---

### Official Review · Reviewer_92iF · 2024-11-03

**Soundness:** 3
**Presentation:** 3
**Contribution:** 3
**Rating:** 5
**Confidence:** 5

**Summary:**

This paper proposes a neural network called InvMSAFold, which takes the protein structure as input, and outputs the parameters of two statistical models. These models are then used to generate a diverse set of protein sequences corresponding to the input structure. By utilizing these simple statistical models, the proposed pipeline effectively addresses two major challenges faced by other inverse-folding methods, such as ESM-IF: (1) the limited diversity of generated sequences and (2) slow sampling speed.

**Strengths:**

In their computational experiments, the authors demonstrated that the sequences generated by their models not only fold into the target structure but also exhibit greater diversity and more effectively capture the correlations between residues at different sites. Furthermore, the showed that this sequence diversity extends to other properties, such as predicted solubility and predicted and predicted thermostability. Overall, this paper represents a new methodological advancement.

**Weaknesses:**

1.	The authors only compare their method with ESM-IF1, and do not compare their method with other state-of-the-art inverse folding methods.
2.	In many places such as in section 1, "ESM-IF" was wrongly typed as "ESM-1F". This may lead readers to perceive the authors as lacking expertise.
3.	The article contains too many grammatical errors.

**Questions:**

1.	The symbols of Eq.(5) is not consistent with that of Eq.(3). It would be better to use consistent symbols.
2.	The proof in section 2.2.1 is incoherent. What is the function of Eq.(5)?
3.	In Eq.(7). It would be better to clarify that Eq.(7) is the L2 regularization term.
4.	In section 3, it would be better to list the number of entries in each dataset.
5.	In section 4.1, what is the necessity of tuning the hyper-parameters of InvMSAFold-AR?
6.	It seems that InvMSAFold-PW performs better than InvMSAFold-AR at larger hamming distance. What is the probable cause?

---

> ### Author Response · Authors · 2024-11-23
>
> We first want to thank the reviewer for carefully reading the manuscript and for the questions raised, which is allowing us to improve the manuscript.
>
> We will address the questions point by point, reporting the original question in italics followed by the answer.
>
> 1) _The authors only compare their method with ESM-IF1, and do not compare their method with other state-of-the-art inverse folding methods_: We thank the reviewer for raising this point, which is allowing us to significantly improve our manuscript. At the time on this answer we are comparing InvMSAFold-AR/PW with [ProteinMPNN](https://www.science.org/doi/10.1126/science.add2187), which will then be added to the manuscript.
> 2) _In many places such as in section 1, "ESM-IF" was wrongly typed as "ESM-1F". This may lead readers to perceive the authors as lacking expertise._: We thank the reviewer for spotting this inaccuracy, which allowed us to polish the manuscript. We diligently re-read the text and correct all typos and especially any possible misspelling if "ESM-IF".
> 3)  _The article contains too many grammatical errors._: We thank the reviewer for raising this point, which is allowing us to polish the manuscript. We are re-reading the manuscript with great care and correcting all grammatical errors to make sure the updated manuscript will not have any.
> 4)   _The symbols of Eq.(5) is not consistent with that of Eq.(3). It would be better to use consistent symbols._: We thank the reviewer for raising this point, as it allowed to improve the clarity of notation in these equations. In Eq.(5) we are replacing all occurrencies of
> $p(\sigma_p| \sigma_{\setminus p})$ with $p^{pw}(\sigma_p| \sigma_{\setminus p} H, J)$ to have consistent symbols with Eq.(3).
> We are, however, not sure if the parts changed correspond to what the reviewer pointed to, so it would be kind if the reviewer could confirm or be more precise.
> 5) _The proof in section 2.2.1 is incoherent. What is the function of Eq.(5)?_: We thank the reviewer for raising this point, which is allowing us to clarify a critical point of the proof mentioned. The role of Eq.(5) is to show that the computation of the different elements of Eq.(3) can be carried efficiently, as the numerator of Eq.(5) is shared for all terms in Eq.(3). The rest of the proof then hence shows how one can compute efficiently the terms in the denominator. We are making sure to better highlight the role of Eq.(5) in the updated version of the manuscript.
> 6) _In Eq.(7). It would be better to clarify that Eq.(7) is the L2 regularization term_: We will make sure to clarify this fact in the updated manuscript.
> 7) _In section 3, it would be better to list the number of entries in each dataset._: We have added this information regarding the entries of each dataset in the updated manuscript.
> 8) _In section 4.1, what is the necessity of tuning the hyper-parameters of InvMSAFold-AR_:    We tuned those parameters because we felt that they played a crucial role in the performance of the model, and hence we wanted to find the combination of them that yielded the most accurate and efficient results. It might seem that we only did this for InvMSAFold-AR, which would completely justify the question of the reviewer, we report in Appendix A.1.2 we also performed such a tuning for InvMSAFold-PW, yet we found that then in many application the tuned model slightly underperformed the original model we were previously using, and therefore we kept that one. We believe this could be due to the fact that the pseudo-likelihood can be not well correlated with generative properties of the model.
> 9) _It seems that InvMSAFold-PW performs better than InvMSAFold-AR at larger hamming distance. What is the probable cause?_:     We thank the reviewer for raising this point, which we also noted and gives us the opportunity to discuss our view on this matter. We do not have a strong explanation for this phenomenon and believe that a more in- depth analysis is needed. However, we speculate that this is linked with what we observe in Figure 7 of the manuscript, i.e. that InvMSAFold-PW generates sequences at higher hamming distances that InvMSAFold-AR, hence it is more accurate in that region which is more on the tail for InvMSAFold-AR.

---

> > ### Author Response · Authors · 2024-12-02
> > **Feedback on updated manuscript**
> >
> > Given that today is the deadline to receive comments, we wanted to know if the reviewer could kindly let us know if he feels our updated manuscript has addressed the comments/questions he raised.
> >
> > If that is the case, we would like to know if the reviewer feels that the changes and additions have raised the quality of the paper. Otherwise, if the reviewer has any other questions regarding the updated manuscript, we would love to have the chance to answer them promptly.

---

### Official Review · Reviewer_ynEz · 2024-11-03

**Soundness:** 3
**Presentation:** 3
**Contribution:** 4
**Rating:** 8
**Confidence:** 3

**Summary:**

The paper presents InvMSAFold, an inverse folding model that generates the parameters of a probability distribution over the space of protein sequences with pairwise interwise interactions, allowing for efficient generation of diverse protein sequences while preserving structural and functional integrity. InvMSAFold is a neural network in which the inputs are the structure backbone coordinates X and the outputs are the parameters of a lightweight sequence model. The lightweight sequence model parameters are used to sample amino acid sequences compatible with the input structure. Training is based on the CATH database, which classifies protein domains into superfamilies and further into clusters based on sequence homology. The model is fast and has uses in protein design and virtual screening. Biologically, the model captures amino acid covariances observed in Multiple Sequence Alignments (MSA) of homologous proteins. The model expands the scope of inverse folding to retrieve a landscape of homologous proteins with similar folds (they say the 'entire' landscape, I don't think they have shown this).  I am overall very enthusiastic about this work.

**Strengths:**

The sampling speed of InvMSAFold is a lot faster than ESM-1F or ProteinMPNN, this is important when you want to generate millions of models, as I think could be reasonable for virtual screening/protein design applications.

InvMSAFold seems able to sample more diverse regions of potential protein structure/function space than ESM-1F, again this is important when you are trying to select for particular properties (substrate specificity, thermostability).

That InvMSAFold is able to capture residue covariances in MSAs may also be useful for better backbone modeling that particular functions could then be engineered into.

**Weaknesses:**

There is not a specific example taken through to the conclusion that the model preserves "structural and functional integrity".  Functional integrity is what you want when you're designing new proteins/doing virtual screening. The authors should consider including such an example or clarifying this statement since that is a major claim of their paper.

I was not clear on the InvMSAFold-AR/-PW. I understand that PW requires MCMC and AR does not but I wonder are there cases/tasks in which a PW vs AR model is more appropriate?

**Questions:**

I was not clear on the InvMSAFold-AR/-PW. I understand that PW requires MCMC sampling and AR does not but I wonder are there cases/tasks in which one of the two PW/AR models is more appropriate?

What would be an example in which you could demonstrate preserved functional integrity that is not directly related to structural integrity in your model's generation of diverse protein sequences?  It seems an important question because when you want to design a protein to do some specific function (bind some small molecule or interact with another protein) you only care about structure to the extent that it acts as a proxy for function. But maybe it doesn't have to be? Do you think your models could get at function outside the restraint of the specific structure that is your input?

---

> ### Author Response · Authors · 2024-11-23
>
> We first want to thank the reviewer for carefully reading the manuscript and for the questions raised, which is allowing us to improve the manuscript.
>
> We will address the questions point by point, reporting the original question in _italics_ followed by the answer.
>
> 1) _There is not a specific example taken through to the conclusion that the model preserves "structural and functional integrity". Functional integrity is what you want when you're designing new proteins/doing virtual screening. The authors should consider including such an example or clarifying this statement since that is a major claim of their paper_:     We thank the reviewer for raising this crucial point. While direct wetlab validation of the generated sequences is not in the scope of our work, we believe that the structural consistency results provide some evidence that functionality is conserved, given the tight relationship between structure and function. We agree, however, that this point should be addressed in a wetlab setting in the future.
> 2) _It was not clear on the InvMSAFold-AR/-PW. I understand that PW requires MCMC and AR does not but I wonder are there cases/tasks in which a PW vs AR model is more appropriate?_:     We thank the reviewer for pointing out this possible source of confusion. It has been observed in a different context [Trinquier, J. et al. (2021)](https://www.nature.com/articles/s41467-021-25756-4) that both models tend to result in very similar performance on different tasks when trained directly on MSAs. However, we found in our paper that the autoregressive model is outperforming the pairwise model when generated by a neural network in our setting. Nonetheless, the pairwise model has a stronger theoretical foundation and has been validated as a good model for protein sequence variability over several decades of research. We strongly suspect that this is due to the fact that the autoregressive model allows for an exact computation of the likelihood and does not need pseudo-likelihoods for training. We therefore strongly feel that this is a result of the training procedure  As a result, this situation could change by more exploring other training procedures in the literature of pairwise models [Barrat-Charlaix, P (2018)](https://www.researchgate.net/publication/342171650_Understanding_and_improving_statistical_models_of_protein_sequences).
> 3) _What would be an example in which you could demonstrate preserved functional integrity that is not directly related to structural integrity in your model's generation of diverse protein sequences? It seems an important question because when you want to design a protein to do some specific function (bind some small molecule or interact with another protein) you only care about structure to the extent that it acts as a proxy for function. But maybe it doesn't have to be? Do you think your models could get at function outside the restraint of the specific structure that is your input?_: We thank the reviewer for raising this crucial point, similar to the question 1 reported in this answer. We refer to the answer there.

---

> > ### Comment · Reviewer_ynEz · 2024-12-02
> > **thanks for your response**
> >
> > I have no additional comments, I assume the authors will update the paper with these responses and I do not need to see the manuscript again.

---

### Official Review · Reviewer_Wqzy · 2024-11-03

**Soundness:** 3
**Presentation:** 3
**Contribution:** 4
**Rating:** 8
**Confidence:** 3

**Summary:**

InvMSAFold is an inverse folding method that is optimized for diversity and speed. The general idea is to use a neural net to predict from an input structure and sequence a pairwise interaction model (a Potts model or Boltzmann machine) that captures the structure-sequence relationship and can be used to efficiently generate sequences that differ largely from the input sequence. To tame the number of parameters (fields and pairwise couplings), InvMSAFold predicts a low-rank approximation of the coupling matrix. The paper proposes two models: 1) InvMSAFold-PW is a full pairwise model that reduces the number of parameters significantly and also allows for efficient learning by using a maximum pseudo-likelihood. A drawback is that sequence generation requires MCMC. 2) InvMSAFold-AR is an autoregressive model whose likelihood is tractable thereby allowing for Maximum-likelihood parameter estimation as well as sampling of sequences in a straight forward fashion. Using various metrics the authors show that InvMSAFold, and in particular InvMSAFold-AR, outperforms current state of the art.

**Strengths:**

* An interesting idea to approach the inverse folding problem (i.e. the problem of generating sequences that fold into a given structure).
* Proposes a low-rank approximation of the couplings and fields of the lightweight sequence model.
* Fast generation of sequences that fit a well to a given structure.

**Weaknesses:**

* The idea of generating a Potts model has already been proposed by Li et al. (2023).

**Questions:**

* How is sampling of InvMSAFold-PW achieved? Which MCMC algorithm do you use?
* By using a PCA projection you show that sequences generated by InvFoldMSA have a better coverage of sequence space. But why do you restrict the analysis to the first two principal components?
* Have you tried AlphaFold3 to validate the sequences generated by InvFoldMSA?

__Typos / grammar__

* Line 117: "and whos outputs"
* Line 212: "can be reduce to"
* Line 248: "robsutly"
* Line 277: "chose" - should be present tense
* Line 302/303: What do you mean by "consistent with the hardness reasoning behind the split"
* Line 475: "becoming worse that both"
* The use of the symbol $\\propto$ to indicate quality up to an additive constant is a bit unusual.

---

> ### Author Response · Authors · 2024-11-23
>
> We first want to thank the reviewer for carefully reading the manuscript and for the questions raised, which is allowing us to improve the manuscript.
>
> We will address the questions point by point, reporting the original question in _italics_ followed by the answer.
>
> 1)  _How is sampling of InvMSAFold-PW achieved? Which MCMC algorithm do you use?_:  We thank the reviewer for raising this question, which allowed to clarify a point on which we had been too vague. Sampling for InvMSAFold-PW is achieved by leveraging the [bmDCA](https://github.com/ranganathanlab/bmdca) sampling library. Such a library implements a standard Monte Carlo algorithm with Metropolis-Hastings proposal as default, yet more advanced proposals as _z-sqrt_ and _z-barker_ from [Livingstone S. & Zanella G. (2019)](https://arxiv.org/abs/1908.11812) are also available. The main advantage of such a library is the computational efficiency of its implementation and parallelization features. We will highlight more these aspects in the updated manuscript.
> 2) _By using a PCA projection you show that sequences generated by InvFoldMSA have a better coverage of sequence space. But why do you restrict the analysis to the first two principal components?_:   We thank the reviewer for raising this point, which gives us the chance to explain better our experimental procedure, especially the connection between plots which might seem independent from one another. The choice of using only the first two principal components was driven by different considerations. First of all, for visualization the first two components are most convenient. Moreover, the choice of reporting the first two PC components is consistent with other relevant works on Potts models in the literature as [Trinquier, J et al (2021)](https://www.nature.com/articles/s41467-021-25756-4). Lastly, the PCA plot as Figure 6 in the manuscript have to be interpreted in conjunction, and not independently, from Figure 5. Indeed the latter, by computing the correlations between the synthetic and true covariances, gives a global result, while the former allows for a local and clear interpretation of consequences of the results of Figure 5. In turn, Figure 5 ensures that the results observed in Figure 6 are not restricted to the first two principal components. We will underline this connection between Figure 5 and 6 in the updated manuscript.
> 3) _Have you tried AlphaFold3 to validate the sequences generated by InvFoldMSA?_:     We agree with the reviewer that such an experiment would be very interesting, unfortunately currently we did not run AF3 on the generated sequences. For the scale of our experiments, we need several thousand forward passes which we were not able to do with the web server available during the conception of our work. The code and weights of AF3 were only released very recently and we do not have sufficient time to run it locally.
> 4) We further thank the reviewer for reporting the typos/grammar above, which allowed us to polish the manuscript.

---

### Author Response · Authors · 2024-11-26
**Revised version of the manuscript**

Thanking again all the reviewers for their fair and interesting points, we have uploaded a revised version of the manuscript. All the questions raised by them should be addressed within the flow of the updated manuscript, hopefully to their liking. In particular, we have added to some experiments also ProteinMPNN as a comparison, as it was suggested by multiple reviewers. Results are reported in Appendix A.1 and B.2 as indicated in the manuscript.

---

### Meta-Review · Area_Chair_UVaw · 2024-12-18

**Metareview:**

The paper presents InvMSAFold, an inverse folding method designed to generate diverse protein sequences from a given structure.

The reviews identified strengths in the speed of the method and the greater diversity that more effectively capture the correlations between residues at different sites.

Weaknesses include limited context including prior work and the fact that the authors compare only to ESM-IF1, and do not compare their method with other state-of-the-art inverse folding methods or to alphafold3.

The authors acknowledge the lack of comparisons and promise to add InvMSAFold-AR/PW with ProteinMPNN to a revised manuscript. Alphafold3 was not available at the scale needed to refold the sequences as suggested. The authors offered to correct typos.

**Additional Comments On Reviewer Discussion:**

The authors addressed the concerns of the reviewers in the discussion and the reviewers who responded considered the updates minor revisions. In response to the author rebuttal some reviewers increased their rating.

---

### Decision · Program_Chairs · 2025-01-22

Accept (Spotlight)